# Soluble amyloid-β precursor peptide does not regulate GABA_B receptor activity

Pascal Dominic Rem[1], Vita Sereikaite[2], Diego Fernández-Fernández[1], Sebastian Reinartz[1], Daniel Ulrich[1], Thorsten Fritzius[1], Luca Trovo[1], Salomé Roux[3], Ziyang Chen[2], Philippe Rondard[3], Jean-Philippe Pin[3], Jochen Schwenk[4], Bernd Fakler[4,5,6], Martin Gassmann[1], Tania Rinaldi Barkat[1], Kristian Strømgaard[2], Bernhard Bettler[1]*

[1]Department of Biomedicine, Pharmazentrum, University of Basel, Basel, Switzerland; [2]Center for Biopharmaceuticals, Department of Drug Design and Pharmacology, University of Copenhagen, Universitetsparken, Copenhagen, Denmark; [3]Institut de Génomique Fonctionnelle, Université de Montpellier, Montpellier, France; [4]Institute of Physiology, Faculty of Medicine, University of Freiburg, Freiburg, Germany; [5]CIBSS Center for Integrative Biological Signalling Studies, University of Freiburg, Freiburg, Germany; [6]Center for Basics in NeuroModulation, Freiburg, Germany

*For correspondence: bernhard.bettler@unibas.ch

**Abstract** Amyloid-β precursor protein (APP) regulates neuronal activity through the release of secreted APP (sAPP) acting at cell surface receptors. APP and sAPP were reported to bind to the extracellular sushi domain 1 (SD1) of GABA_B receptors (GBRs). A 17 amino acid peptide (APP17) derived from APP was sufficient for SD1 binding and shown to mimic the inhibitory effect of sAPP on neurotransmitter release and neuronal activity. The functional effects of APP17 and sAPP were similar to those of the GBR agonist baclofen and blocked by a GBR antagonist. These experiments led to the proposal that sAPP activates GBRs to exert its neuronal effects. However, whether APP17 and sAPP influence classical GBR signaling pathways in heterologous cells was not analyzed. Here, we confirm that APP17 binds to GBRs with nanomolar affinity. However, biochemical and electro-physiological experiments indicate that APP17 does not influence GBR activity in heterologous cells. Moreover, APP17 did not regulate synaptic GBR localization, GBR-activated K⁺ currents, neurotransmitter release, or neuronal activity in vitro or in vivo. Our results show that APP17 is not a functional GBR ligand and indicate that sAPP exerts its neuronal effects through receptors other than GBRs.

## Editor's evaluation

The manuscript is very relevant to the field of Alzheimer's disease and the signal transduction mechanisms of GABA_B receptors. It is a rigorous study that addresses the physiological function of peptide isoforms of APP vis-a-vie G protein-coupled GABA_B receptors. The manuscript presents strong experimental evidence of this interaction but also concludes that although an APP peptide, APP17, binds a GABA_B receptor, there is a lack of clearly demonstrable functional effects.

## Introduction

Amyloid precursor protein (APP or A4 protein) is a transmembrane protein that undergoes proteolytic processing by secretases. The amyloidogenic pathway generates amyloid-β peptides (Aβ) that are key etiological agents of Alzheimer's disease (AD). The competing non-amyloidogenic pathway generates secreted APP (sAPP) variants that modulate spine density, synaptic transmission, plasticity processes, and rescue synaptic deficits in *App^-/-* mice (*Müller et al., 2017*; *Haass and Willem, 2019*; *Tang, 2019*).

It is assumed that cell surface receptors mediate the synaptic effects of sAPP (*Richter et al., 2018*; *Haass and Willem, 2019*; *Tang, 2019*; *Barthet and Mulle, 2020*). It was recently proposed that sAPP acts at G protein-coupled GABA$_B$ receptors (GBRs) to modulate synaptic transmission and neuronal activity (*Rice et al., 2019*) (reviewed by *Haass and Willem, 2019*; *Korte, 2019*; *Tang, 2019*; *Yates, 2019*; *Barthet and Mulle, 2020*). GBRs are good candidates for mediating the functional effects of sAPP because they regulate neurotransmitter release, neuronal inhibition, and synaptic plasticity processes by reducing cAMP levels and gating Ca$^{2+}$ and K$^+$ channels (*Lüscher and Slesinger, 2010*; *Gassmann and Bettler, 2012*; *Pin and Bettler, 2016*; *Barthet and Mulle, 2020*).

GBRs are composed of a GB1a or GB1b subunit with a GB2 subunit, which generates heterodimeric GB1a/2 and GB1b/2 receptors (*Pin and Bettler, 2016*). GB1a differs from GB1b by the presence of two N-terminal sushi domains (SD1/2). GB1a/2 and GB1b/2 receptors predominantly localize to pre- and postsynaptic sites, respectively (*Vigot et al., 2006*). APP and sAPP bind to SD1 of GB1a (*Schwenk et al., 2016*; *Dinamarca et al., 2019*; *Rice et al., 2019*). Synthetic APP peptides of 9 or 17 amino acid residues, termed APP9 and APP17, are sufficient for binding and inducing a stable conformation in SD1 (*Rice et al., 2019*; *Feng et al., 2021*; *Yang et al., 2022*). APP, sAPP, and APP17 bind to recombinant SD1 with a K$_D$ of 183, 431, and 810 nM, respectively (*Dinamarca et al., 2019*; *Rice et al., 2019*). APP17 and sAPP reduced the frequency of miniature excitatory postsynaptic currents (mEPSCs) in brain slices, thus mimicking the activity of the orthosteric GBR agonist baclofen at presynaptic GB1a/2 receptors (*Rice et al., 2019*). The antagonist CGP55845 reduced the inhibitory effect of APP17 on the mEPSC frequency, in further support of APP17 activating GBRs (*Rice et al., 2019*). Moreover, APP17 inhibited neuronal activity in the hippocampus of anesthetized mice (*Rice et al., 2019*), consistent with a GBR-mediated inhibition of glutamate release and/or activation of postsynaptic K$^+$ currents. Based on these experiments, Rice and colleagues proposed that APP17 and sAPP are functional GBR ligands (*Rice et al., 2019*). However, they presented no evidence that APP17 or sAPP activate classical GBR-activated G protein signaling pathways, which is necessary to demonstrate a direct action at GBRs. In separate studies, the binding of APP to GB1a/2 receptors in cis was shown to mediate receptor transport to presynaptic sites and to stabilize receptors at the cell surface (*Hannan et al., 2012*; *Dinamarca et al., 2019*). Accordingly, *App$^{-/-}$* mice exhibited a 75% decrease of axonal GBRs in hippocampal neurons, which significantly reduced GBR-mediated presynaptic inhibition (*Dinamarca et al., 2019*), as already shown in earlier experiments (*Seabrook et al., 1999*). However, recombinant sAPP had no effect on GBR-mediated G protein activation in transfected HEK293 cells (*Dinamarca et al., 2019*). Thus, it remains controversial whether APP17 and sAPP are functional ligands at GBRs or not.

The reported effects of APP17 on synaptic release and neuronal activity (*Rice et al., 2019*) suggest that APP17 acts as a positive allosteric modulator (PAM) or ago-PAM (PAM with agonistic properties) at GBRs. APP17 could also increase constitutive activity of GBRs (*Grünewald et al., 2002*; *Rajalu et al., 2015*) by binding to SD1. To clarify whether APP17 influences GBR activity, we studied the effects of APP17 on classical GBR signaling pathways in transfected HEK293T cells, cultured neurons, acute hippocampal slices, and in anesthetized mice. Our experiments confirm that APP17 binds with nanomolar affinity to purified recombinant SD1/2 protein and to GB1a/2 receptors expressed in HEK293T cells. However, in our hands, APP17 neither induced conformational changes at GBRs consistent with receptor activation nor influenced GBR-mediated G protein activity, cAMP inhibition or Kir3-type K$^+$ currents. APP17 also failed to modulate constitutive GBR activity in the absence or presence of APP expressed in cis or in trans. Likewise, APP17 did not influence K$^+$ currents or mEPSC frequencies in cultured hippocampal neurons, reduce the amplitude of evoked EPSCs (eEPSCs) in hippocampal slices, or modulate neuronal activity in living mice. Thus, our in vitro and in vivo data indicate that receptors other than GBRs mediate the synaptic effects of sAPP.

## Results

### APP17 binds to purified recombinant SD1/2 protein and GB1a/2 receptors expressed in HEK293T cells

We purchased APP17 and scrambled sc-APP17 peptides from the same commercial provider as Rice and colleagues (*Rice et al., 2019*; *Figure 1A*). For displacement experiments, we additionally synthesized fluorescent APP17-TMR and sc-APP17-TMR peptides labeled with TAMRA (5 (6)-carboxytetra

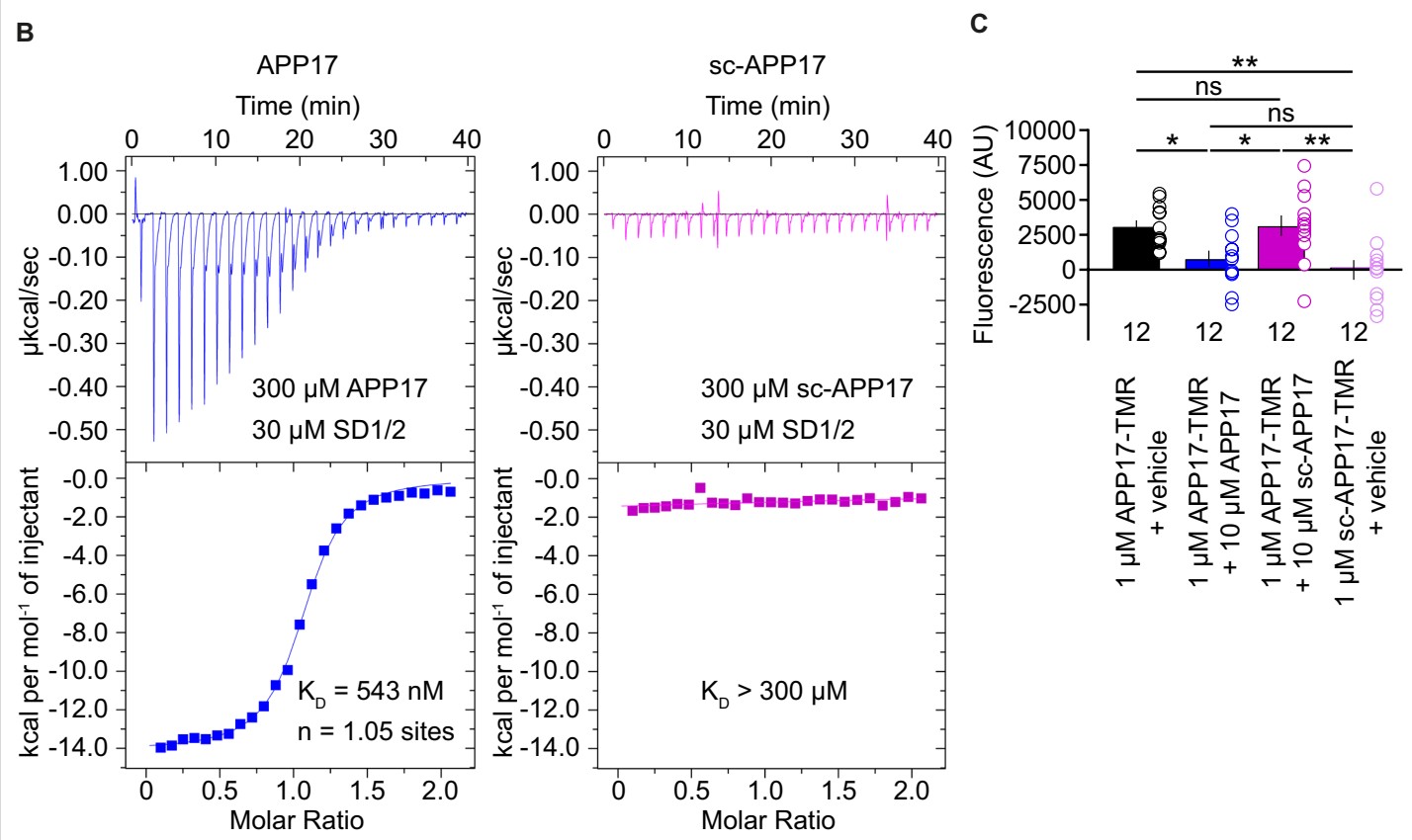

**Figure 1.** Characterization of synthetic APP17 and sc-APP17 peptides. (**A**) Sequence alignment of APP17, sc-APP17, APP17-TMR, and sc-APP17-TMR peptides. Residues critical for sushi domain 1 (SD1) binding are shown in red. (**B**) Representative isothermal titration calorimetry (ITC) diagrams of the titrations of SD1/2 protein in solution (30 µm) with APP17 (blue) or sc-APP17 (magenta) (300 µm in the syringe); raw heat signature (top) and integrated molar heat release (bottom). The calculated stoichiometry of APP17:SD1/2 protein is 1.05, the $K_D$ 543 nM. sc-APP17 showed no binding to SD1/2 protein. (**C**) Bar graph showing APP17-TMR (1 µM) binding to GB1a/2 receptors in HEK293T cells in the presence of vehicle (black), 10 µM APP17 (blue), and 10 µM sc-APP17 (magenta). sc-APP17-TMR (1 µM) served as a negative control. The background fluorescence of sc-APP17-TMR (1 µM) at HEK293T cells transfected with empty vector was subtracted. Data are means ± SEM. The number of independent experiments is indicated. ns = not significant, *p<0.05, **p<0.01, one-way ANOVA with Holm-Sidak's multiple comparisons test. Source file containing ICT and TMR fluorescence data is available in *Figure 1—source data 1*.

The online version of this article includes the following source data for figure 1:

**Source data 1.** Characterization of synthetic APP17 peptides.

methylrhodamine; *Figure 1A*). ESI-LC-MS and RP-UPLC analysis revealed that all peptides had the expected molecular weight and purity (*Figure 1A*). Isothermal titration calorimetry (ITC) showed that APP17 interacts with purified recombinant SD1/2 protein (*Schwenk et al., 2016*) in solution with an ~1:1 stoichiometry and a $K_D$ of 543 nM (*Figure 1B*). This agrees well with the published $K_D$ of 810 nM for binding of APP17 to recombinant SD1 (*Rice et al., 2019*). In contrast, sc-APP17 showed

no detectable binding to SD1/2 ($K_D$ >300 µM). APP17-TMR exhibited significantly more binding to HEK293T cells expressing GB1a/2 receptors than sc-APP17-TMR (*Figure 1C*). Accordingly, 10 µM APP17 but not sc-APP17 displaced 1 µM APP17-TMR from GB1a/2 receptors (*Figure 1C*). In all subsequent experiments, we used the commercial APP17 peptide validated for binding to recombinant SD1/2 protein and GB1a/2 receptors in HEK293T cells.

## APP17 does not induce the active state of GB1a/2 receptors

Upon binding of APP17, SD1 adopts a stable conformation (*Rice et al., 2019*) that possibly influences GBR activity allosterically. We used a fluorescence resonance energy transfer (FRET)-based conformational sensor in transfected HEK293T cells to analyze whether APP17 induces the inter-subunit rearrangement associated with GBR activation (*Lecat-Guillet et al., 2017*). The FRET sensor is based on GB1a and GB2 subunits fused with ACP and SNAP (*Figure 2A*), respectively. These tags are enzymatically modified with time-resolved FRET compatible fluorophores (HA-GB1a-ACP with CoA-Lumi4-TB [Donor], Flag-SNAP-GB2 with SNAP-RED [Acceptor]). This FRET sensor discriminates between GBR agonists with different efficacies and between PAMs with distinct modes of action (*Lecat-Guillet et al., 2017*). For FRET experiments, we used APP17 and sc-APP17 at 1 and 10 µM. These concentrations are above the $K_D$ of APP17 at SD1 and comparable to those used in previous functional experiments (25 nM to 5 µM) (*Rice et al., 2019*). As expected, GABA decreased FRET in a dose-dependent manner (*Figure 2B*). APP17 at 10 µM, when applied alone (basal, *Figure 2B*), significantly increased instead of decreased FRET (*Figure 2B and C*). This FRET increase can be rationalized in two ways. First, since GBRs exhibit constitutive activity (*Grünewald et al., 2002*; *Rajalu et al., 2015*), there is an equilibrium between active and inactive states of the receptor. Constitutive activity decreases basal FRET because a fraction of receptors is in the active state. Ligands stabilizing the inactive conformation will increase basal FRET. Therefore, APP17 is potentially an inverse agonist of GBRs. However, since all our other functional experiments reveal no inverse agonistic properties (see below), APP17 binding to SD1 likely influences the positioning of the SNAP-tag located on top of GB2, thereby decreasing the mean distance between the fluorophores and increasing FRET efficacy. Of note, APP17 lacks allosteric properties as it did not significantly alter GABA potency (*Figure 2D*). In summary, the FRET conformational sensor provides no evidence for agonistic or PAM activity of APP17.

## APP17 does not influence G protein activation by GB1a/2 receptors

We directly tested whether APP17 influences G protein activation by GBRs in transfected HEK293T cells. We used a bioluminescent resonance energy transfer (BRET) assay monitoring dissociation of Gα from Gβγ upon receptor activation (*Turecek et al., 2014*; *Figure 3A*). Application of GABA to cells expressing GB1a/2 together with Gαo-RLuc, Venus-Gγ2, and Gβ2 lead to a BRET decrease between Gαo-RLuc and Venus-Gγ2 (*Figure 3A*). Subsequent blockade of GB1a/2 receptors by the inverse agonist CGP54626 (*Grünewald et al., 2002*) increased BRET due to re-association of G protein subunits (*Figure 3A*). Of note, CGP54626 increased BRET above baseline, consistent with substantial constitutive activity of GBRs (*Grünewald et al., 2002*; *Rajalu et al., 2015*). Accordingly, application of CGP54626 alone to transfected HEK293T cells increased BRET (*Figure 3A*). Application of 10 µM GABA did not overcome receptor inhibition by 10 µM CGP54626 (*Figure 3A*). The presence of APP695 in cis did not alter constitutive activity of the receptor (*Figure 3A*). APP17 and sc-APP17 at 1 or 10 µM did not significantly influence receptor activity while subsequent GABA application to the same cells induced the expected BRET decrease (*Figure 3B*). GABA-induced BRET decreases were similar in the presence of APP17 or sc-APP17 (*Figure 3B*). These experiments indicate that APP17 at 1 or 10 µM exerts no agonistic, inverse agonistic, antagonistic, or allosteric properties at GB1a/2 receptors. Moreover, application of APP17 or sc-APP17 to HEK293T cells expressing GB1a/2 receptors and APP695 in cis or in trans had no effect on GBR activity (*Figure 3C and D*). Subsequent application of GABA was equally effective in decreasing BRET in the presence of APP17 or sc-APP17 (*Figure 3C and D*). These experiments showed that APP17 does not influence GBR-mediated G protein activation in the absence or presence of APP695.

## APP17 does not influence GB1a/2 receptor-mediated Gαi signaling

Measuring Gαi signaling provides another means to study possible functional effects of APP17 on GBR activity. We analyzed whether APP17 influences GB1a/2-mediated Gαi signaling using an assay

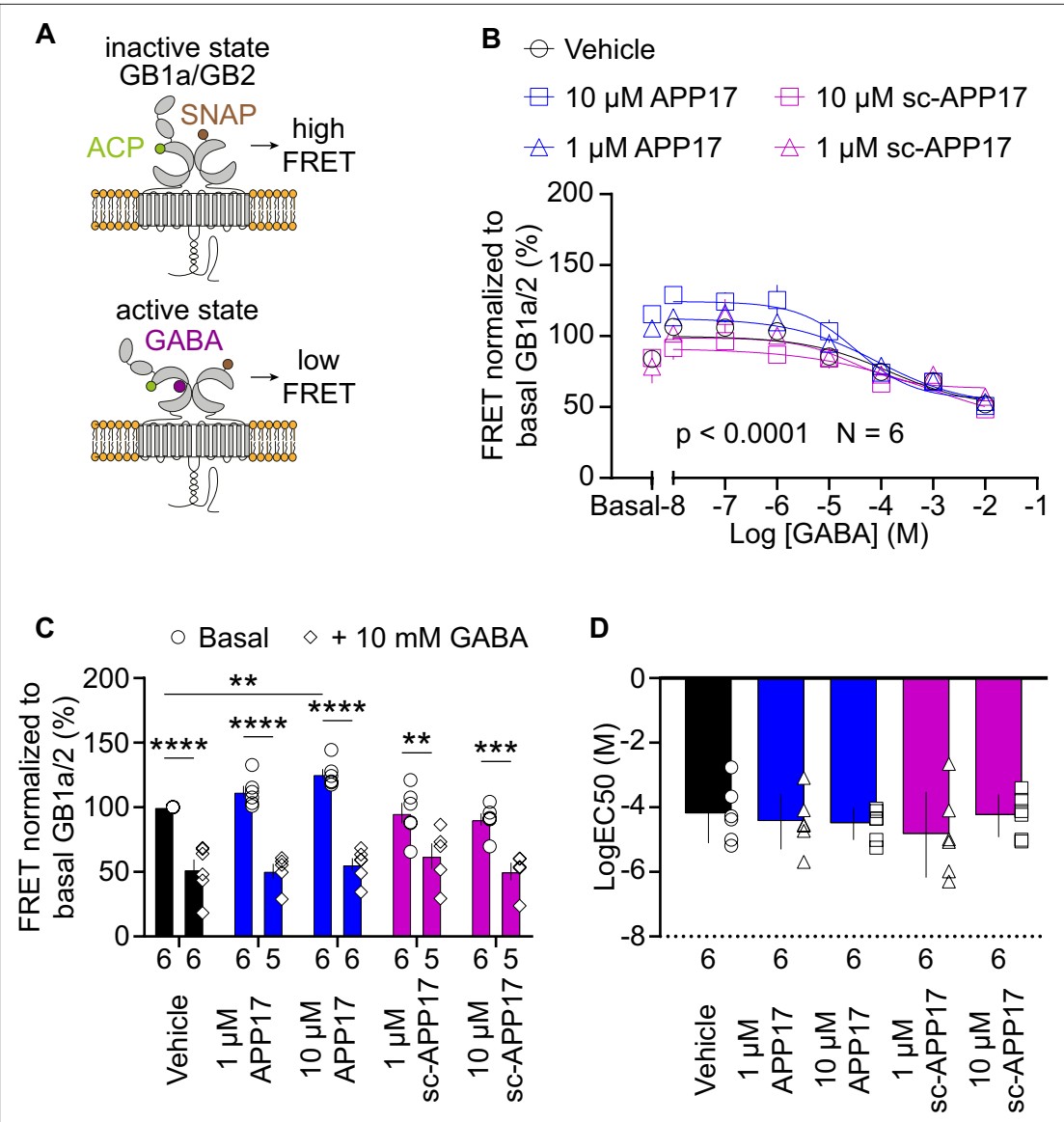

**Figure 2.** APP17 does not induce the active state of GB1a/2 receptors. (**A**) Assay measuring inter-subunit fluorescence resonance energy transfer (FRET) between fluorophore labeled ACP and SNAP tags in the GB1a and GB2 subunits, respectively. In the absence of receptor agonists, the ACP and SNAP tags are in close proximity resulting in high FRET. Activation of GB1a/2 receptors induces a conformational change in the extracellular domains leading to a reduction in FRET. (**B**) GABA dose-response curves in the presence of APP17 (blue) or sc-APP17 (magenta) at 1 μM (triangles) and 10 μM (squares) or vehicle (black) exhibit significant differences. (**C**) Bar graphs showing FRET in the presence of 1 or 10 μM APP17 (blue) and sc-APP17 (magenta) or vehicle (black). Under basal conditions (circles), the presence of 10 μM APP17 resulted in a significant increase of FRET, whereas no significant changes in FRET were observed for all other conditions when compared to vehicle. In the presence of 10 mM GABA (diamonds) no significant differences in FRET were detected with APP17 or sc-APP17 at 1 or 10 μM compared to vehicle. In all conditions, the presence of 10 mM GABA induced a significant reduction in FRET compared to basal. (**D**) LogEC$_{50}$ values of individual GABA dose-response curves exhibit no significant differences between conditions. Data are means ± SEM. Three outliers were identified in (C) using the ROUT method (PRISM) with Q=1% (source file). The number of independent experiments is indicated. **p<0.01, ***p<0.001, ****p<0.0001, two-way ANOVA with Sidak's multiple comparisons test. Source file containing FRET data is available in *Figure 2—source data 1*.

The online version of this article includes the following source data for figure 2:

**Source data 1.** FRET analysis of GBR conformational changes.

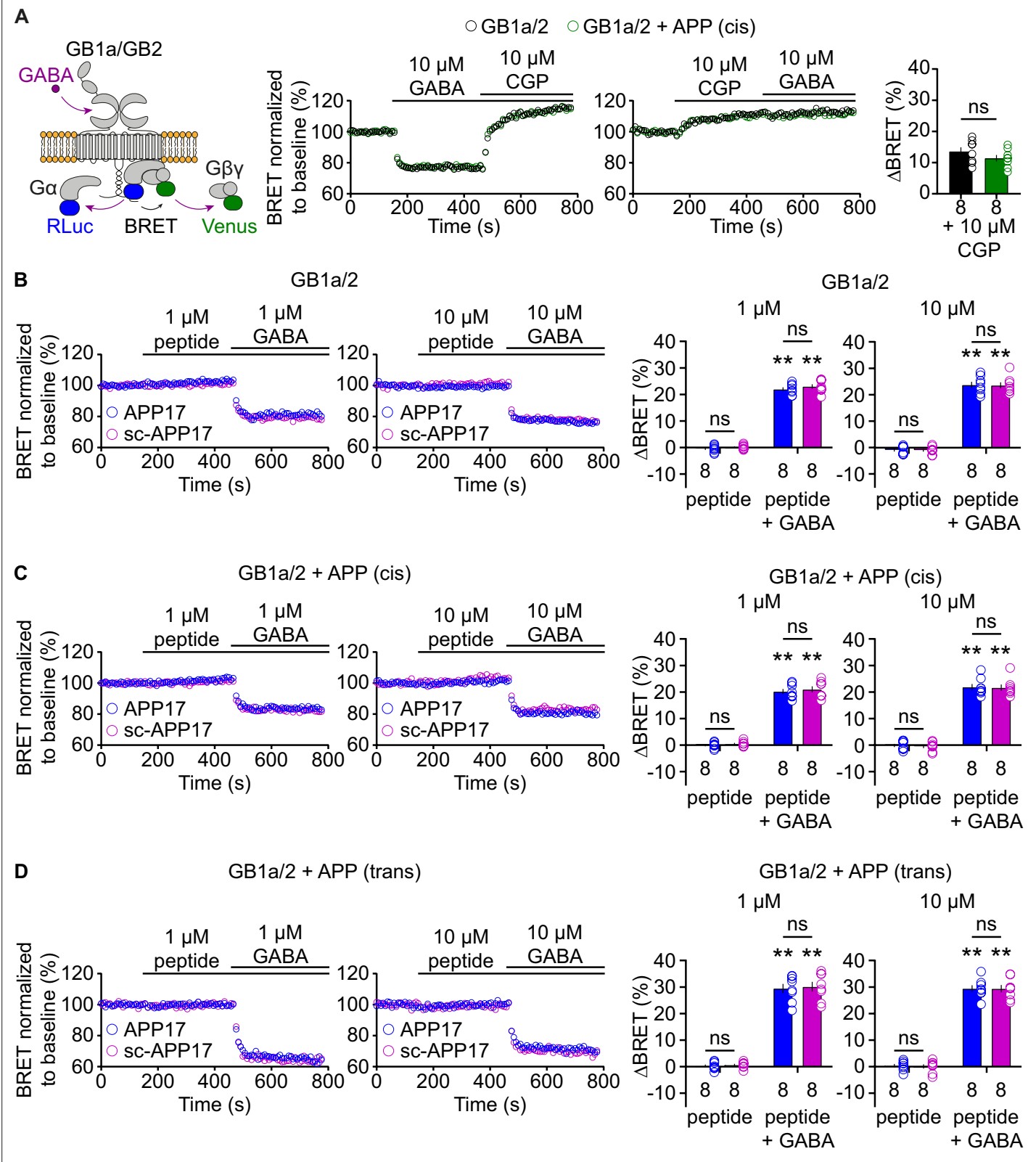

**Figure 3.** APP17 is not an agonist, inverse agonist, antagonist, or allosteric modulator at GB1a/2 receptors expressed in HEK293T cells in a bioluminescent resonance energy transfer (BRET) assay monitoring G protein activation. (**A**) *Left:* Assay measuring BRET between Gαo-RLuc and Venus-Gγ2. GB1a/2 receptor activation leads to dissociation of the heterotrimeric G protein and a consequent decrease in BRET. *Right:* Individual experiments showing GABA-induced BRET changes at GB1a/2 receptors. The inverse agonist CGP54626 reverses GABA-induced BRET changes above baseline,

*Figure 3 continued on next page*

*Figure 3 continued*

indicating constitutive GB1a/2 receptor activity. Likewise, direct application of CGP54626 increased BRET levels above baseline. Subsequent application of GABA did not overcome receptor inhibition. Bar graphs summarize CGP54626-induced BRET changes. Note that application of CGP54626 resulted in similar inhibition of constitutive GB1a/2 receptor activity in the absence (black) or presence (green) of APP695 in cis. (**B**) Neither APP17 (blue) nor sc-APP17 (magenta) at 1 µM (left) or 10 µM (right) altered BRET in cells expressing GB1a/2 receptors. In the same cells, GABA at 1 µM (left) and 10 µM (right) induced the expected decrease in BRET. The GABA-induced BRET change is similar in the presence of APP17 and scAPP17, indicating the absence of allosteric properties of the peptides at GB1a/2 receptors. Bar graphs summarize BRET changes determined in experiments as shown to the left. (**C,D**) Neither APP17 (blue) nor sc-APP17 (magenta) at 1 µM (left) or 10 µM (right) altered BRET in cells expressing GB1a/2 receptors together with APP695 in cis (**C**) or in trans (**D**). Bar graphs summarize BRET changes. Data are means ± SEM. The number of independent experiments is indicated in the bar graphs. ns = not significant, two-way ANOVA with Sidak's multiple comparisons test. **p<0.01, one-sample Wilcoxon test (non-parametric) against 0. Source file containing BRET data is available in *Figure 3—source data 1*.

The online version of this article includes the following source data for figure 3:

**Source data 1.** BRET analysis of GBR-mediated G protein activation.

monitoring cAMP-dependent protein kinase A (PKA) activity in transfected HEK293T cells. This assay is based on regulatory and catalytic PKA subunits tagged with the N- or C-terminal fragments of RLuc (R-RLuc-N, C-RLuc-C) (*Stefan et al., 2007*; *Figure 4A*). GB1a/2 receptor activation by 10 µM GABA inhibits adenylyl cyclase, which inactivates PKA and increases luminescence due to association of R-RLuc-N with C-RLuc-C (*Figure 4A*). Blockade of GB1a/2 receptors with 10 µM CGP54626 decreased luminescence below baseline, again revealing substantial constitutive receptor activity in this assay system (*Figure 4A*). The presence of APP695 in cis did not alter receptor activity (*Figure 4A*). APP17 or sc-APP17 at 10 µM exhibited no agonistic, inverse agonistic, or antagonistic properties at GB1a/2 receptors (*Figure 4B*). GABA-mediated PKA inactivation was comparable in the presence of APP17 or sc-APP17, again supporting that APP17 does not act as a PAM (*Figure 4B*). Moreover, APP17 or sc-APP17 did not significantly alter GB1/2 receptor activity in the presence of APP695 in cis (*Figure 4C*).

It is conceivable that the APP17 concentrations used are not optimal for detecting functional effects at recombinant GB1a/2 receptors. Therefore, we determined APP17 dose-response curves using an accumulation assay based on artificially coupling GB1a/2 receptors via chimeric Gαqi to phospholipase C (PLC) (*Conklin et al., 1993*; *Figure 5A*). PLC activity was monitored with a serum responsive element-luciferase (SRE-luciferase) reporter amplifying the receptor response (*Cediel et al., 2022*). Increasing concentrations of GABA yielded similar sigmoidal dose-response curves in the absence and presence of APP695 expressed in cis or in trans (*Figure 5—figure supplement 1*), showing that binding of APP695 does not influence receptor activity. APP17 or sc-APP17 lacked agonistic properties at concentrations up to 100 µM, the highest concentration tested (*Figure 5A*). CGP54626 blocked constitutive and GABA-induced receptor activity (*Figure 5B*). APP17 or sc-APP17 at concentrations of 1 and 10 µM did not influence constitutive GB1a/2 receptor activity (*Figure 5B*). Pre-incubation with 1 or 10 µM of APP17 or sc-APP17 did not significantly influence the GABA dose-response curve in the absence (*Figure 5C*) or presence of APP695 expressed in cis (*Figure 5D*), corroborating that the peptides lack agonistic, PAM, inverse agonistic, or antagonistic properties.

## APP17 does not bias GB1a/2 signaling toward Gαq

GBRs are classical Gαo/i-coupled receptors that normally lack the ability to mobilize intracellular Ca$^{2+}$ (*Pin and Bettler, 2016*). However, it has been proposed that in some neurons GBRs increase intracellular Ca$^{2+}$ by activating Gαq and PLC (*Karls and Mynlieff, 2015*). It is possible that APP17 is a biased ligand preferentially stabilizing a conformational state of GBRs that elicits signaling through Gαq (*Slosky et al., 2021*). APP17 did not activate PLC in the SRE-luciferase assay (*Figure 5*). However, high levels of exogenous Gαqi in this assay may interfere with signaling through endogenous Gαq. We therefore analyzed APP17 activity at GB1a/2 receptors in a Ca$^{2+}$ mobilization assay (*Werthmann et al., 2021*). HEK293T cells expressing GB1a/2 or control mGlu5 receptors were loaded with the Ca$^{2+}$ indicator Calcium 6 and stimulated with 1 or 10 µM APP17, GABA, or (S)-3,5-DHPG (*Figure 5—figure supplement 2*). Application of APP17 or GABA did not induce significant Ca$^{2+}$ mobilization in cells expressing GB1a/2 receptors (*Figure 5—figure supplement 2A and B*). Expression of GB1a/2 receptors in HEK2093T cells was confirmed by immunostaining for GB1 and GB2 (*Figure 5—figure supplement 2C*). Application of the selective group I mGlu receptor agonist DHPG to cells expressing mGlu5 receptors induced a concentration-dependent increase in intracellular Ca$^{2+}$ (*Figure 5—figure*

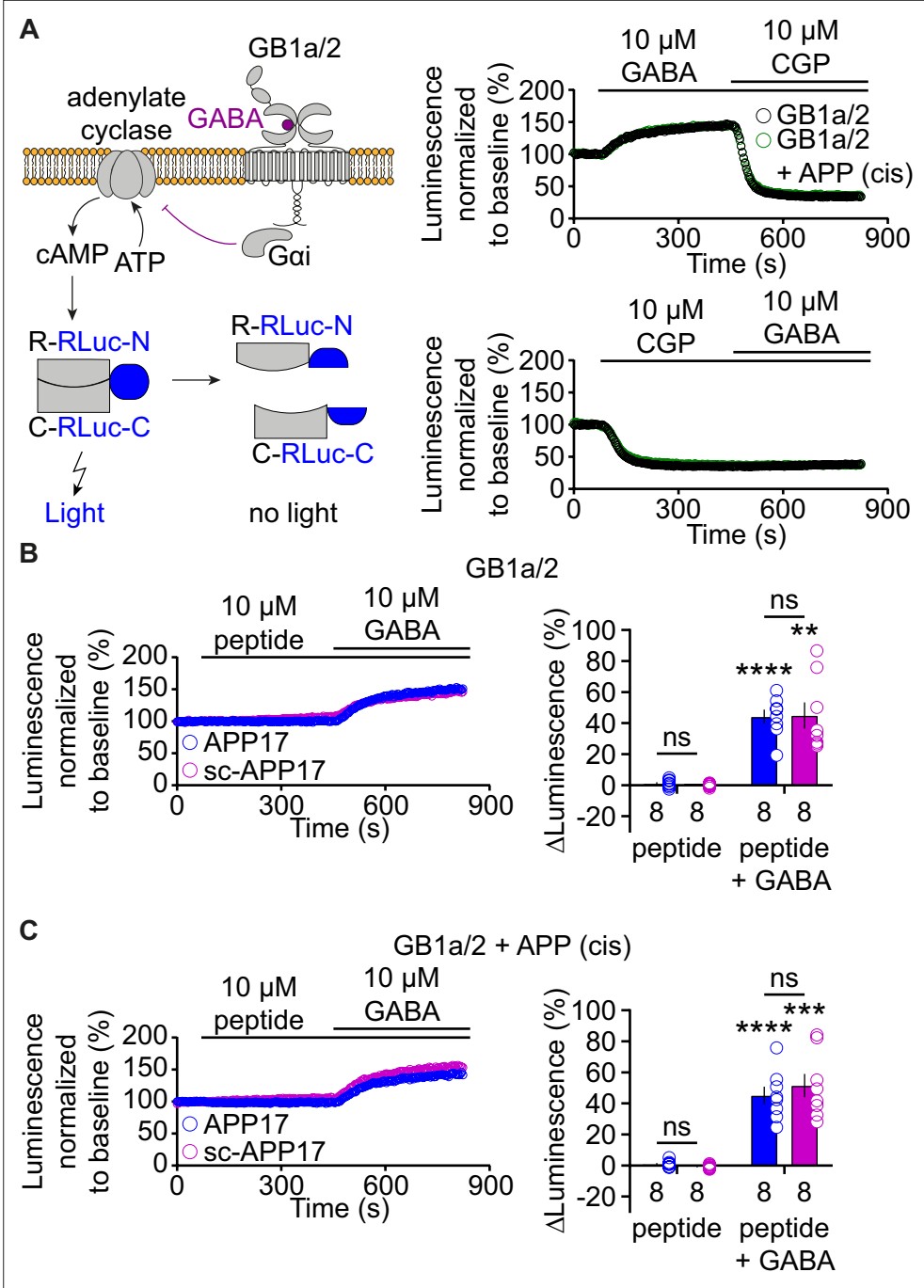

**Figure 4.** APP17 is not an agonist, inverse agonist, antagonist, or allosteric modulator at GB1a/2 receptors expressed in HEK293T cells in an assay monitoring Gαi signaling. (**A**) *Left:* Assay monitoring dissociation of the regulatory (R) and catalytic (C) subunits of the tetrameric protein kinase A (PKA) holoenzyme upon cAMP binding. PKA subunits were tagged with N- or C-terminal fragments of RLuc (R-RLuc-N, C-RLuc-C). GB1a/2 receptor activation by GABA reduces intracellular cAMP levels, promotes reconstitution of RLuc activity, and increases luminescence. *Right:* Individual experiments showing GABA-induced luminescence changes. Blockade of GB1a/2 receptor activity with CGP54626 decreased luminescence below baseline, indicating constitutive GB1a/2 receptor activity. Subsequent application of GABA did not overcome receptor inhibition. (**B**) Neither APP17 (blue) nor sc-APP17 (magenta) altered luminescence in HEK293T cells expressing GB1a/2 receptors. In the same cells, GABA induced the expected luminescence increases. GABA-induced luminescence changes are similar in the presence of APP17 and sc-APP17. Bar graphs summarize the luminescence changes. (**C**) Neither APP17 (blue) nor sc-APP17 (magenta) induced luminescence changes in HEK293T cells expressing GB1a/2 receptors together with APP695

*Figure 4 continued on next page*

*Figure 4 continued*

in cis. Application of GABA to the same cells resulted in the expected luminescence increases. GABA-induced luminescence changes are similar in the presence of APP17 or sc-APP17. Bar graphs summarize the luminescence changes. Data are means ± SEM. The number of independent experiments is indicated. ns = not significant, two-way ANOVA with Sidak's multiple comparison test. **p<0.01, ***p<0.001, ****p<0.0001, one-sample t-test against 0. Source file containing PKA luminescence data is available in *Figure 4—source data 1*.

The online version of this article includes the following source data for figure 4:

**Source data 1.** Luminescence analysis of GBR-mediated Gαi singaling.

supplement 2A and B*), as expected (*Werthmann et al., 2021*). These experiments indicate that APP17 does not elicit Gαq signaling at GB1a/2 receptors in HEK293T cells.

## APP17 does not influence synaptic GBR localization

It is conceivable that prolonged exposure of cultured hippocampal neurons to APP17 affects synaptic GBR levels, for example, by stabilizing GB1a/2 receptors at the cell surface or displacing APP from GB1a/2 receptors (*Hannan et al., 2012*; *Dinamarca et al., 2019*). Exposure of cultured hippocampal neurons to 5 μM APP17 or sc-APP17 for 1 hr did not result in a significant difference in GB2 protein levels at glutamatergic synapses, which were identified by vGluT1 and PSD95 immunostaining (*Figure 6*). The functional effects of APP17 previously observed in cultured neurons (*Rice et al., 2019*) are therefore not caused by altered GBR levels at synapses.

## APP17 does not influence [$^{35}$S]GTPγS binding in brain tissue

Native GBRs form multi-protein complexes with auxiliary proteins (*Pin and Bettler, 2016*; *Schwenk et al., 2016*). It is conceivable that the functional APP17 effects observed in neurons (*Rice et al., 2019*) depend on the presence of GBR-associated proteins that are missing in heterologous expression systems. Binding of the non-hydrolysable GTP analog [$^{35}$S]guanosine-5'-O-(3-thio)triphosphate ([$^{35}$S]GTPγS) to Gαi/o in brain membranes allows to quantify G protein activation by native GBRs (*Galvez et al., 2000*; *Schuler et al., 2001*). GABA dose-response curves for native GBRs in the absence and presence of 1 μM APP17 did not significantly differ from each other and exhibited similar EC$_{50}$ and Emax values (*Figure 7*). This finding supports that 1 μM APP17 has no agonistic, inverse agonistic, antagonistic, or allosteric effects at native GBRs.

## APP17 does not influence GBR-activated K$^+$ currents in neurons and transfected HEK293T cells

APP17 signaling through native GBRs may depend on protein-protein interactions that are not preserved in the membrane preparations used for the [$^{35}$S]GTPγS binding assay. Therefore, we tested whether APP17 influences GBR-mediated Gβγ signaling to K$^+$ channels in cultured hippocampal neurons using patch-clamp electrophysiology, which preserves the native environment of receptors (*Schuler et al., 2001*; *Vigot et al., 2006*). Application of 5 μM APP17 or sc-APP17 to hippocampal neurons did not elicit any currents, in contrast to the same concentration of baclofen (*Figure 8A and B*). Co-application of APP17 or sc-APP17 with baclofen elicited currents of similar amplitudes as baclofen alone (*Figure 8B and C*), indicating that APP17 exerts no allosteric properties. APP17 or sc-APP17 also did not trigger K$^+$ currents in transfected HEK293T cells expressing Kir3 channels, nor did the peptides alter K$^+$ currents in the presence of 5 μM GABA (*Figure 8—figure supplement 1*). These findings support that APP17 has no agonistic, PAM, or antagonistic effects at GBR-activated K$^+$ currents.

## APP17 does not influence synaptic transmission in hippocampal neurons

GB1a/2 receptors are abundant at axon terminals where they inhibit neurotransmitter release (*Vigot et al., 2006*). Acute exposure of cultured hippocampal neurons to APP17 at 250 nM was shown to reduce the frequency of mEPSCs (*Rice et al., 2019*), consistent with an activation of presynaptic GB1a/2 receptors. A GBR antagonist blocked the effect of APP17 on the mEPSC frequency, supporting a GBR-dependent mechanism. Since our experiments showed no functional effects of APP17 at GBRs,

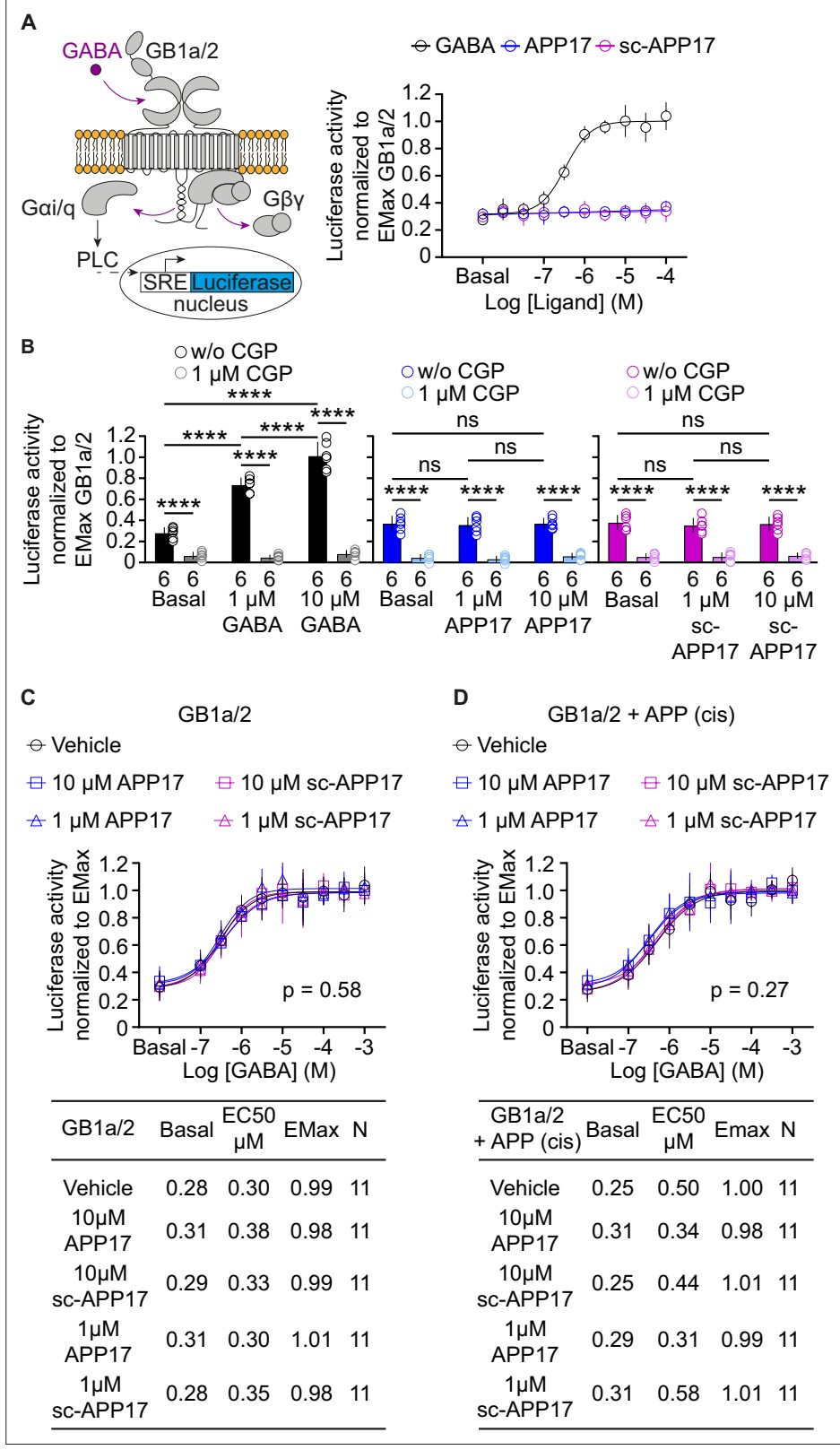

**Figure 5.** APP17 is not an agonist, inverse agonist, allosteric modulator, or antagonist at GB1a/2 receptors expressed in HEK293T cells when monitoring $Ga_{qi}$ signaling in an accumulation assay. (**A**) *Left:* Assay monitoring phospholipase C (PLC)-dependent FLuc expression under control of the serum response element (SRE). GB1a/2 receptors were artificially coupled to PLC by stably expressing the chimeric G protein subunit $G\alpha qi$. GB1a/2

*Figure 5 continued on next page*

*Figure 5 continued*

receptors and SRE-FLuc reporter were transiently expressed in HEK293T-Gαqi cells. *Right:* Dose-response curve showing that GABA (black) but not APP17 (blue) or sc-APP17 (magenta) induces FLuc activity in transfected cells. (**B**) CGP54626 blocked constitutive and GABA-induced FLuc activity in transfected cells. Constitutive GB1a/2 receptor activity is unchanged in the presence of APP17 (middle) or sc-APP17 (right) at 1 or 10 μM, indicating the absence of inverse agonist properties of the peptides at GB1a/2 receptors. (**C,D**) APP17 (blue) or sc-APP17 (magenta) at 1 μM (triangles) or 10 μM (squares) did not significantly alter GABA dose-response curves in the absence (**C**) or presence (**D**) of APP695 in cis, indicating that the peptides do not allosterically regulate GB1a/2 receptors. Tables show basal, EC$_{50}$, and Emax values derived from the curve fits. All data are mean ± SD. The number of independent experiments is indicated in the bar graphs or tables. Linear regression curve fit of 6 (APP17, sc-APP17, **A**) independent experiments per condition. Non-linear regression curve fits of 6 (GABA, **A**) or 11 (**C,D**) independent experiments per condition. p=0.58, p=0.27, extra sum-of-squares F test. Source file containing FLuc activity data is available in *Figure 5—source data 1*.

The online version of this article includes the following source data and figure supplement(s) for figure 5:

**Source data 1.** SRE-luciferase analysis of GBR-mediated Gαqi signaling.

**Figure supplement 1.** APP695 expressed in cis or in trans with GB1a/2 receptors exerts no allosteric effects on Gαqi signaling in HEK293T cells.

**Figure supplement 1—source data 1.** SRE-luciferase analysis of GBR-mediated Gαqi signaling in the presence of APP695 in cis or in trans.

**Figure supplement 2.** APP17 does not increase intracellular Ca$^{2+}$ in HEK293T cells expressing GB1a/2 receptors.

**Figure supplement 2—source data 1.** Analysis of GBR and mGlu5 receptor mediated intracellular Ca$^{2+}$ increases.

we sought to replicate the reported decrease in mEPSC frequency after APP17 application to cultured hippocampal neurons (*Rice et al., 2019*). Using the same incubation time and concentration of APP17 (*Rice et al., 2019*), we were unable to detect a significant effect of APP17 on the mEPSC frequency of cultured hippocampal neurons. Subsequent application of baclofen to the same neurons resulted in a significantly reduced mEPSC frequency (*Figure 9A and B*). We additionally studied whether APP17 at 250 nM influences eEPSC amplitudes in acute hippocampal slices. Postsynaptic GBR-activated K$^+$ currents were blocked with a Cs$^+$-based intracellular solution, which allowed to specifically monitor the activity of presynaptic GBRs. We found that baclofen but not APP17 was able to reduce the amplitudes of eEPSCs (*Figure 9C–E*).

## APP17 does not influence spontaneous neuronal activity in the auditory cortex of anesthetized mice

Two-photon Ca$^{2+}$ imaging showed that APP17 suppresses neuronal activity of CA1 pyramidal cells in anesthetized mice (*Rice et al., 2019*). We therefore similarly performed two-photon Ca$^{2+}$ imaging experiments in anesthetized transgenic mice to analyze whether APP17 modulates spontaneous activity in cortical neurons, where the density of GBRs is high (*Bischoff et al., 1999*). We crossed Ai95(RCL-GCaMP6f)-D mice (*Madisen et al., 2015*) with Nex-Cre mice (*Goebbels et al., 2006*) to express the Ca$^{2+}$ indicator GCaMP6f under the Nex-promoter, which allowed us to record Ca$^{2+}$ transients in layer 2/3 neurons of the auditory cortex. APP17, sc-APP17, and baclofen solutions were perfused over the right cortical surface in a fixed sequence (*Figure 10A*). To control for potential time-dependent changes of spontaneous activity under isoflurane anesthesia (*Magnuson et al., 2014*), we perfused artificial cerebrospinal fluid (ACSF) before and after perfusion of sc-APP17 and APP17. The results showed that sc-APP17 and APP17 at concentrations of 5 μM had no significant effect compared to ACSF, even after 60 min of perfusion (*Figure 10D and E*). In contrast, 5 μM of baclofen reduced spontaneous Ca$^{2+}$ transients after 15 min of perfusion. Therefore, we were unable to confirm that APP17 influences neuronal activity in vivo.

## Discussion

Proteolytic processing of APP through the non-amyloidogenic pathway liberates sAPP, which modulates synaptic functions, presumably by acting at neuronal cell surface receptors (*Ishida et al., 1997*; *Bour et al., 2004*; *Taylor et al., 2008*; *Claasen et al., 2009*; *Aydin et al., 2011*; *Hick et al., 2015*; *Müller et al., 2017*; *Richter et al., 2018*). Nanomolar concentrations of sAPP were shown to have

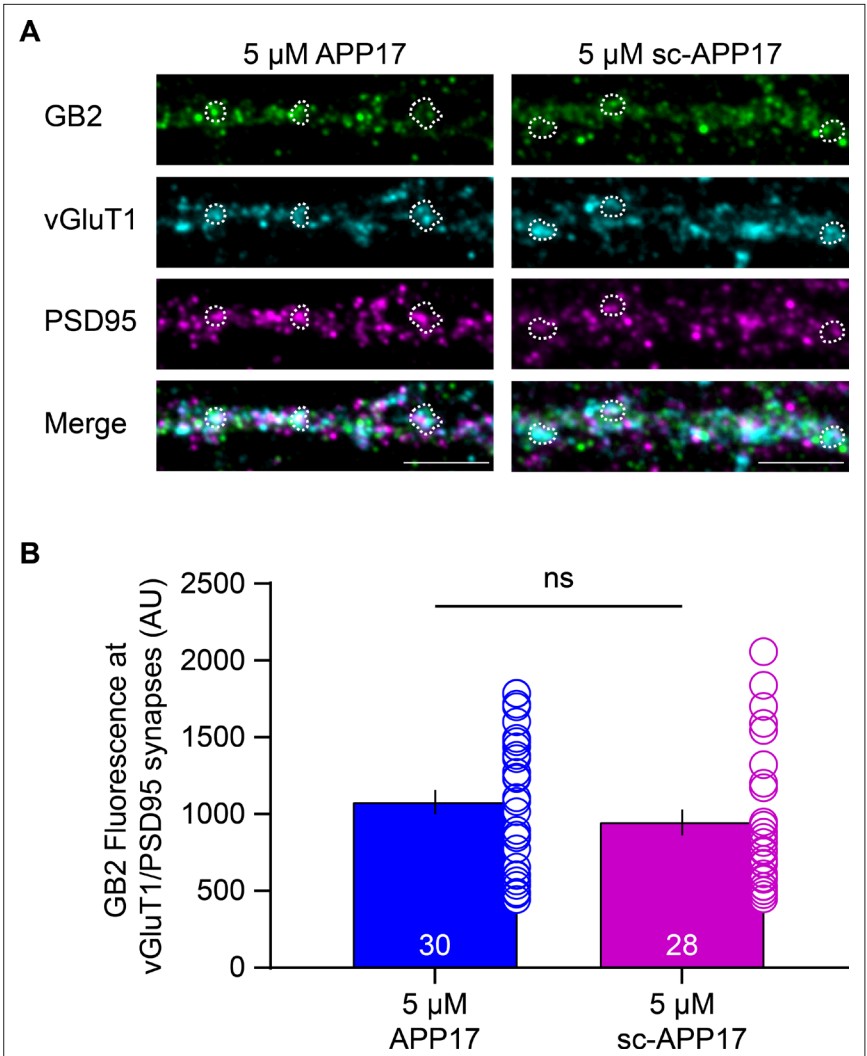

**Figure 6.** APP17 does not influence GBR levels at glutamatergic synapses. (**A**) Images of dendrites of cultured hippocampal neurons (DIV14) exposed to 5 µM APP17 or sc-APP17 for 1 hr. Immunofluorescence labeling of GB2 (Alexa Fluor 488), vGluT1 (Alexa Fluor 647), PSD95 (Alexa Fluor 555), and their overlay (merged) is shown. Dashed areas indicate examples of vGluT1/PSD95 synapses. Scale bar: 5 µM. (**B**) Bar graphs showing GB2 (Alexa Fluor 488) immunofluorescence at vGluT1/PSD95 synapses. Data are mean ± SEM. The number of analyzed images from three independent preparations is indicated. ns = not significant, Mann-Whitney test (non-parametric). Source file containing GB2 immunofluorescence data is available in *Figure 6—source data 1*.

The online version of this article includes the following source data for figure 6:

**Source data 1.** Analysis of synaptic GBR levels.

PAM activity at heterologously expressed α7 nicotinic acetylcholine receptors, suggesting that nicotinic receptors mediate some of the effects of sAPP (*Richter et al., 2018*). Recent experiments identified GB1a/2 receptors as receptors for sAPP (*Rice et al., 2019*). GB1a/2 receptors are predominantly expressed at presynaptic sites, where they regulate neurotransmitter release (*Vigot et al., 2006*). It was shown that sAPP and APP17, a peptide of 17 amino acids corresponding to the SD1 binding site of APP, reduce the frequency of mEPSCs and inhibit neuronal activity (*Rice et al., 2019*). While these findings received much attention and are consistent with activation of GBRs (*Haass and Willem, 2019*; *Korte, 2019*; *Tang, 2019*; *Yates, 2019*), fundamental questions remained. For example, it is unclear how a conformational change in SD1, induced by sAPP or APP17 binding, increases GBR activity. High-resolution structures of the GBR heterodimer in the apo, antagonist-bound, agonist-bound and agonist- and PAM-bound states in complex with the G protein are available and provide

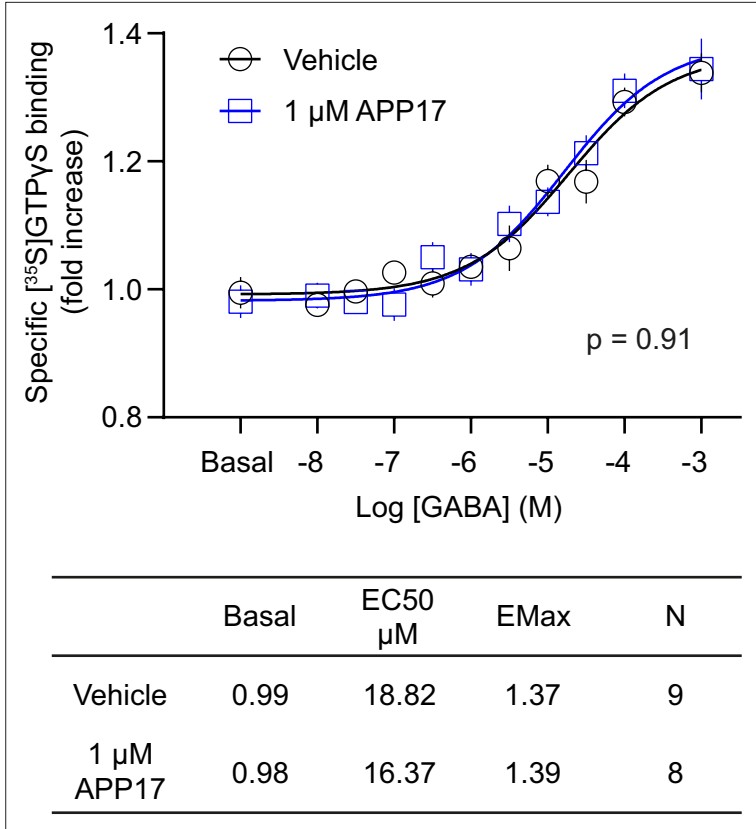

| | Basal | EC50 μM | EMax | N |
|---|---|---|---|---|
| Vehicle | 0.99 | 18.82 | 1.37 | 9 |
| 1 μM APP17 | 0.98 | 16.37 | 1.39 | 8 |

**Figure 7.** APP17 is not an agonist, antagonist, or allosteric modulator at native GB1a/2 receptors in [35S]GTPγS binding experiments. [35S]GTPγS binding in brain membrane preparations of WT mice induced by increasing concentrations of GABA is not altered in the presence of APP17 (blue). The table shows basal, EC50, and Emax values derived from non-linear regression curve fits. Experiments with vehicle and APP17 were performed with membrane preparations from the same mouse. Data are mean ± SEM. Non-linear regression curve fit of nine (vehicle) and eight (APP17) independent experiments with nine different mice. p=0.91, extra sum-of-squares F test. Source file containing [35S]GTPγS data is available in *Figure 7—source data 1*.

The online version of this article includes the following source data for figure 7:

**Source data 1.** [35S]GTPγS analysis of GBR activity in mouse brain membrane preparations.

detailed insights into the activation mechanism of GBRs (*Mao et al., 2020*; *Papasergi-Scott et al., 2020*; *Park et al., 2020*; *Shaye et al., 2020*; *Shaye et al., 2021*; *Shen et al., 2021*). These structures show that the N-terminal SD1 is neither part of the binding sites for orthosteric or allosteric ligands, nor alters pharmacological receptor properties (*Kaupmann et al., 1998*) or participates in receptor activation (*Evenseth et al., 2020*; *Shaye et al., 2021*). Therefore, there is no straightforward explanation for potential functional effects of sAPP or APP17 at GBRs. Moreover, *Rice et al., 2019*, did not analyze whether sAPP or APP17 regulate GB1a/2 receptors in heterologous cells, which is important to demonstrate a direct action at the receptor. In fact, in an earlier report showing interaction of native GBRs with APP, we found no evidence for recombinant sAPP regulating GB1a/2 receptors expressed in heterologous cells (*Dinamarca et al., 2019*). However, native GBRs form receptor complexes with additional proteins (*Pin and Bettler, 2016*; *Schwenk et al., 2016*; *Bettler and Fakler, 2017*) and these proteins could be necessary for the observed effects of APP17 on receptor activity.

The aim of this study was to clarify whether APP17 can activate recombinant and/or native GBRs. We were able to confirm that APP17 binds to purified recombinant SD1/2 protein with a $K_D$ of 543 nM, which is similar to the $K_D$ determined earlier (*Rice et al., 2019*). For functional experiments in HEK293T cells, we used a range of established cell-based GBR assays reporting (1) conformational changes associated with receptor activation, (2) G protein activation, (3) cAMP inhibition, and (4) Kir3 channel activation. In all these assays, APP17 had no agonistic, PAM, or antagonistic properties at GB1/2 receptors. APP17 also did not influence constitutive GBR activity in the presence or absence

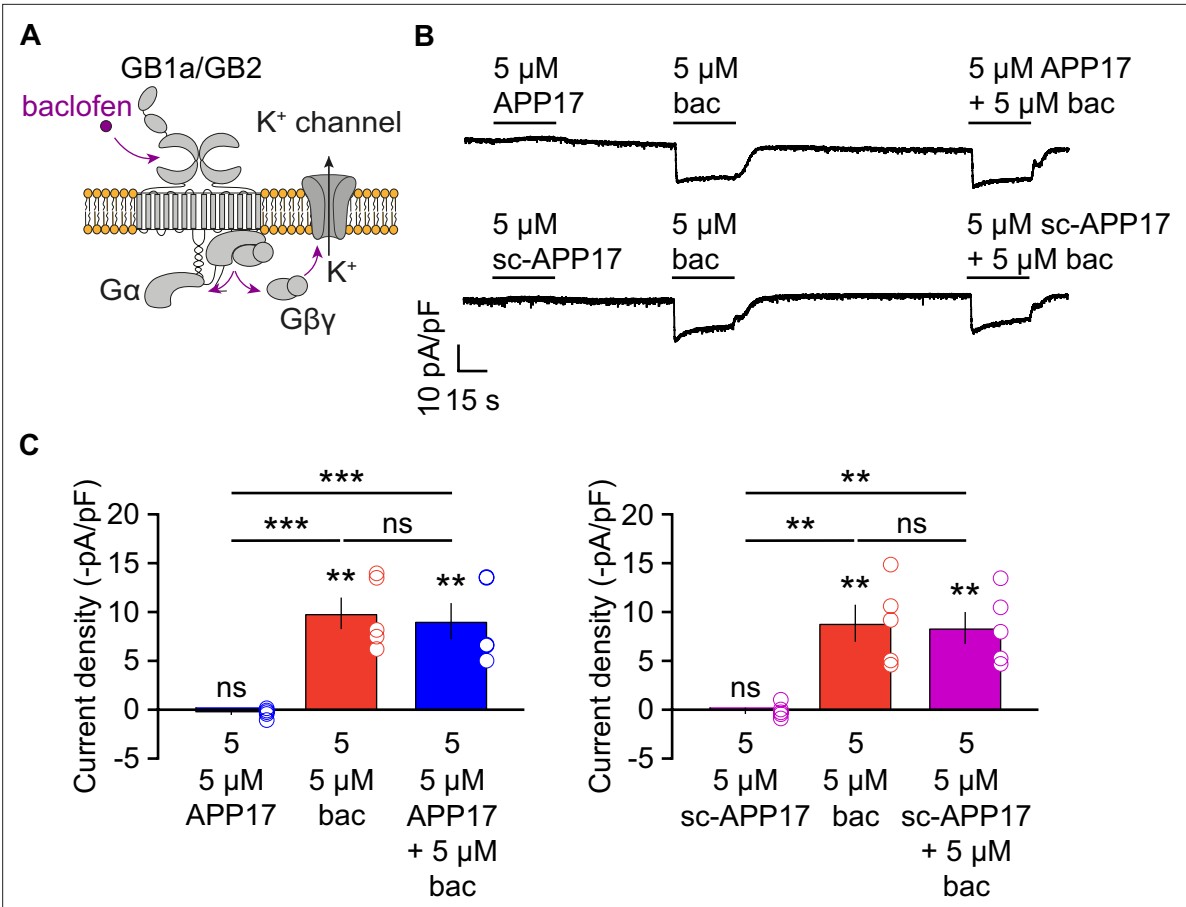

**Figure 8.** APP17 does not evoke or influence GB1a/2 receptor-mediated K⁺ currents in cultured hippocampal neurons. (**A**) GBR activation with baclofen results in the dissociation of the heterotrimeric G protein and the subsequent activation of K⁺ channels by Gβγ. (**B**) Representative traces showing that neither APP17 (top) nor sc-APP17 (bottom) evoke GB1a/2 receptor-induced K⁺ currents in cultured hippocampal neurons. Application of baclofen alone or in the presence of APP17 or sc-APP17 yielded similar current amplitudes, showing that APP17 does not allosterically modulate baclofen-induced currents. (**C**) Bar graphs showing K⁺ current densities determined in experiments as shown to the top. Data are means ± SEM. The number of independent experiments is indicated in the bar graphs. ns = not significant, **p<0.01, ***p<0.001, paired one-way ANOVA with Holm-Sidak's multiple comparisons test (to compare different means) and one-sample t-test against 0. Source file containing K⁺ current data is available in *Figure 8—source data 1*.

The online version of this article includes the following source data and figure supplement(s) for figure 8:

**Source data 1.** Analysis of GBR-mediated K⁺ currents in cultured hippocampal neurons.

**Figure supplement 1.** APP17 does not evoke or influence GB1a/2 receptor-induced Kir3 currents in transfected HEK293T cells.

**Figure supplement 1—source data 1.** Analysis of GBR-mediated Kir3 currents in HEK293T cells.

of APP695, which competes with APP17 for binding at SD1. It is well established that GBRs normally signal through Gαo/i-type G proteins (*Pin and Bettler, 2016*). However, Gαq signaling was proposed as an alternative signaling mode of GBRs in select neuronal populations (*Karls and Mynlieff, 2015*). We therefore considered the possibility that APP17 acts as a biased ligand favoring Gαq over Gαo/i signaling. However, we found no evidence that APP17 mobilizes intracellular Ca²⁺ in transfected HEK293T cells expressing GB1a/2. APP17 also did not regulate native GBRs in experiments assessing (1) G protein activation in brain membranes, (2) activation of K⁺ currents in cultured neurons, (3) mEPSC frequency in cultured neurons, (4) eEPSC amplitude modulation in acute hippocampal slices, and (5) neuronal activity in anesthetized mice. Consistent with a lack of APP17 effects on GBRs, we found no evidence that prolonged application of APP17 alters synaptic GBR levels. Thus, our findings are consistent with a lack of functional effects of APP17 at GB1a/2 receptors, confirming our earlier data with sAPP (*Dinamarca et al., 2019*). The lack of functional effects is not due to a faulty APP17 peptide, since the APP17 peptide used in functional experiments was validated for binding

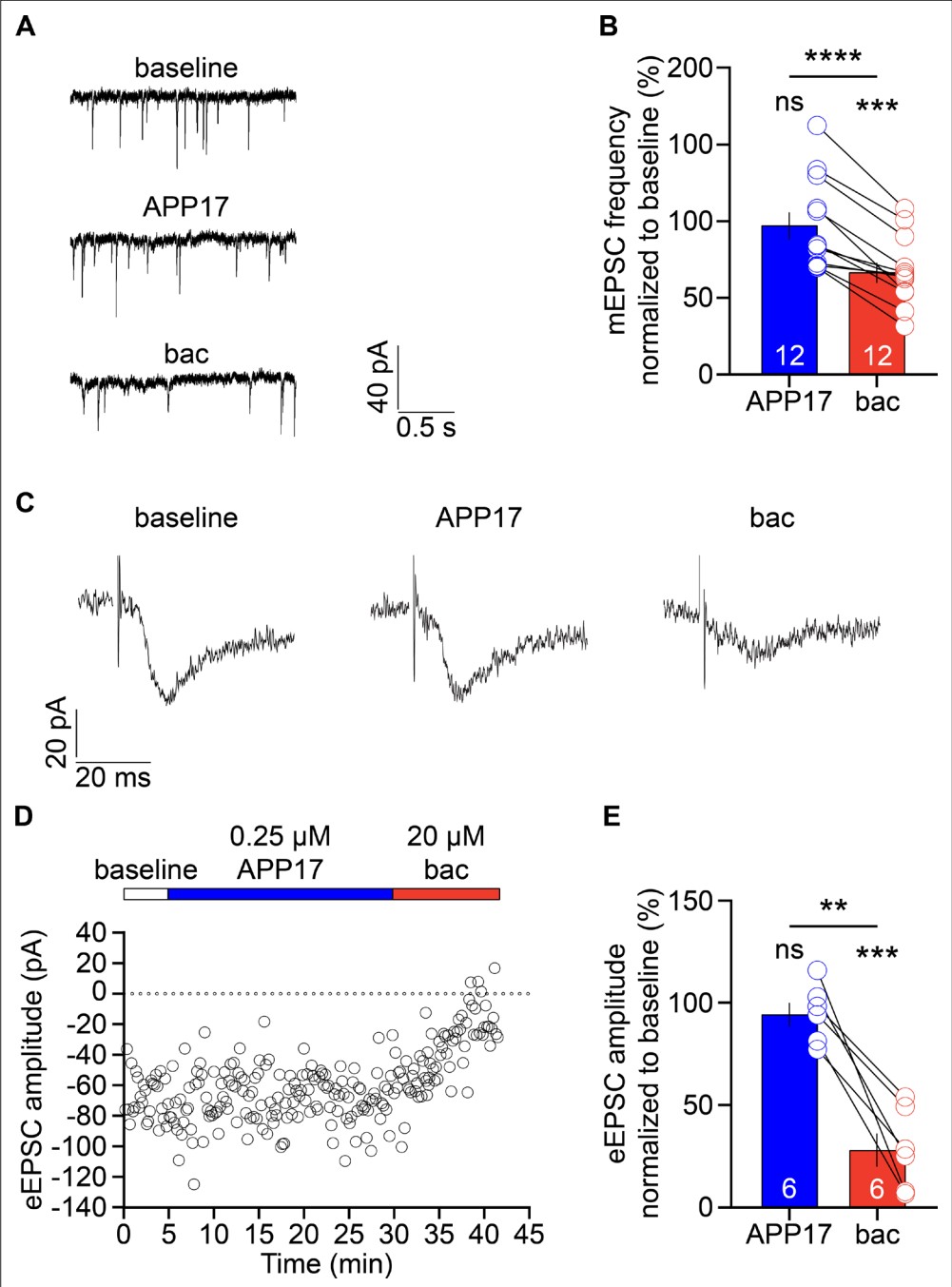

**Figure 9.** APP17 does not influence excitatory synaptic transmission. (**A**) Sample mEPSCs at baseline and in the presence of 0.25 µM APP17 or 30 µM baclofen (bac) recorded from cultured hippocampal neurons (DIV13-16). (**B**) Bar graphs showing the mEPSC frequency normalized to baseline in the presence of APP17 (blue) or baclofen (red). Data are means ± SEM. The number of recorded neurons is indicated. ns = not significant; ****p<0.0001, paired Student's t-test; ***p<0.001, one-sample t-test against 100. (**C**) Sample eEPSCs at baseline and in the presence of APP17 and baclofen recorded in CA1 pyramidal neurons of acute hippocampal slices. (**D**) Time course of eEPSC amplitudes in a CA1 pyramidal neuron. 0.25 µM APP17 and 20 µM baclofen were bath applied as indicated. (**E**) Bar graphs of the EPSC amplitude reduction in the presence of APP17 and baclofen. Data are means ± SEM. The number of recorded neurons from six different mice is indicated. ns = not significant; **p<0.01, paired t-test; ***p<0.001, one sample t-test against 100. Source file containing mEPSC and eEPSC data is available in *Figure 9—source data 1*.

The online version of this article includes the following source data for figure 9:

**Source data 1.** Analysis of mEPSCs and eEPSCs in hippocampal neurons.

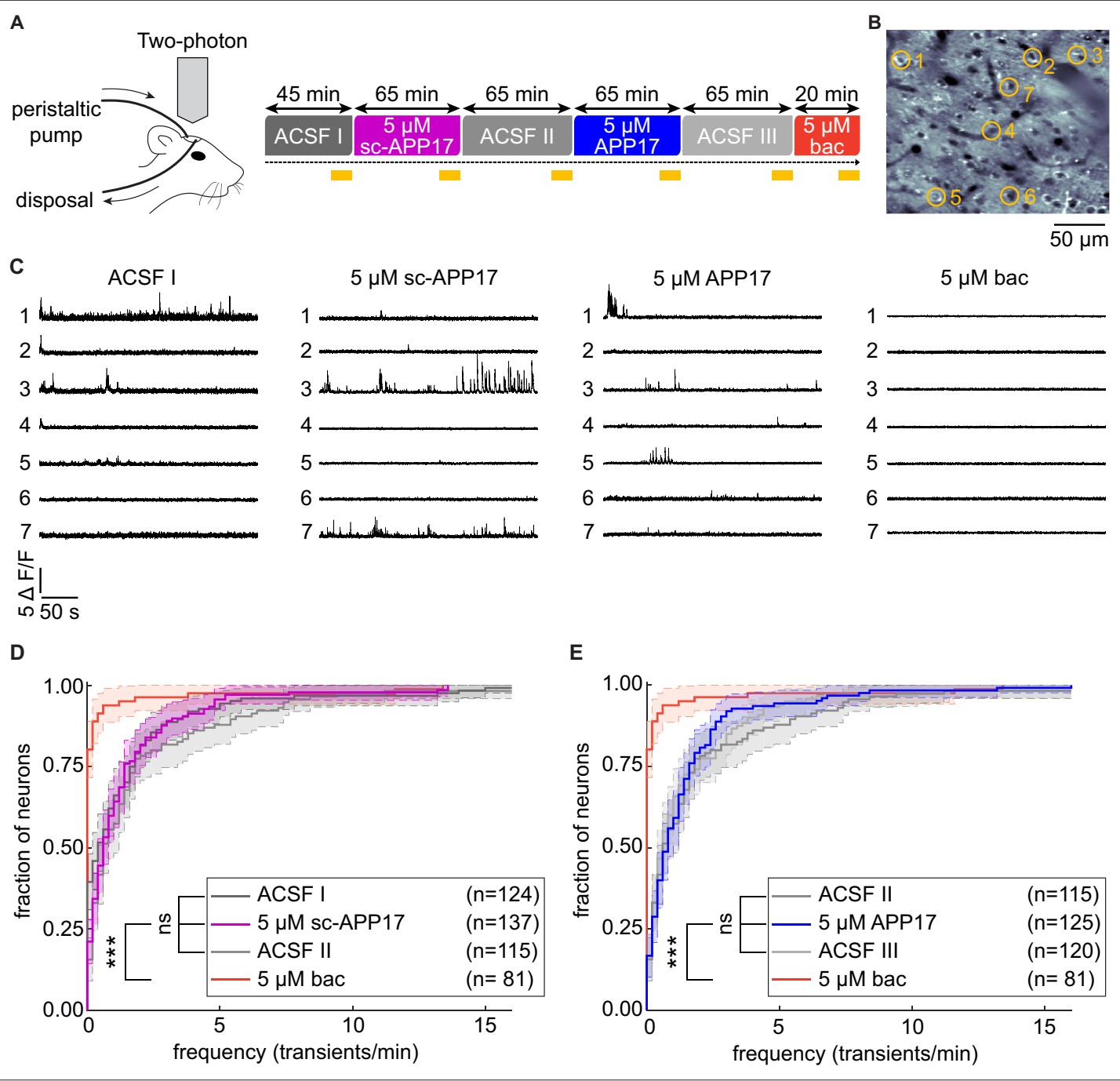

**Figure 10.** APP17 does not influence spontaneous neuronal activity in the auditory cortex of mice. (**A**) *Left:* Two-photon imaging of Ca²⁺ transients in the auditory cortex of anesthetized mice during perfusion of artificial cerebrospinal fluid (ACSF), APP17, sc-APP17, and baclofen. *Right:* Scheme of the experimental design. Time specifications denote the durations of the perfusions. Yellow lines indicate the two-photon Ca²⁺ imaging periods (5 min each). (**B**) In vivo two-photon image of neurons expressing GCaMP6f. Representative neurons selected to illustrate Ca²⁺ transients in (**C**) are marked with yellow circles. (**C**) Ca²⁺ transients of neurons shown in (**B**) across the entire 5 min imaging period of a given condition. (**D**) Cumulative distribution of the frequency of Ca²⁺ transients, comparing sc-APP17 with baseline (ACSF I) and washout (ACSF II) and perfusion with baclofen (bac). (**E**) Cumulative distribution of the frequency of Ca²⁺ transients, comparing APP17 with baseline (ACSF II) and washout (ACSF III) and baclofen. (**D,E**) 95% confidence intervals are shown as shaded areas. The number of neurons recorded in each condition are indicated. Kruskal-Wallis multicomparison test: APP17 versus ACSF I, II, III, and sc-APP17 are not significantly different (p>0.05); bac versus ACSF I, II, III, sc-APP17, and APP17 are all significantly different (p<0.0001). For p-values, see *Figure 10—source data 2*. Source file containing Ca²⁺ transient data is available in *Figure 10—source data 1*.

The online version of this article includes the following source data for figure 10:

*Figure 10 continued on next page*

*Figure 10 continued*

**Source data 1.** Analysis of neuronal activity in anesthetized mice.

**Source data 2.** Analysis of neuronal activity in anesthetized mice: p-values between experimental conditions.

recombinant SD1/2 protein and GB1a/2 receptors expressed in HEK293T cells. In all our experiments, we used GABA or baclofen to control for receptor activity. It therefore appears that sAPP mediates its neuronal effects through receptors other than GBRs. APP binding to GBRs probably mainly evolved to control receptor trafficking in axons and stabilize APP and GB1a/2 receptors at the cell surface (*Hannan et al., 2012*; *Dinamarca et al., 2019*). As a final remark it is important to note that the concentration of the abundant sAPPα variant in the interstitial fluid reaches ~1 nM (*Dobrowolska et al., 2014*). Considering a $K_D$ of 183 nM for the APP interaction with GB1a (*Dinamarca et al., 2019*), it is unlikely that endogenous levels of sAPP would reach concentrations high enough to directly activate the receptor or displace APP from GB1a/2 receptors.

# Materials and methods
## Plasmids and reagents
The following plasmids were used: Flag-GB1a (*Adelfinger et al., 2014*); Flag-GB2, APP695 (*Dinamarca et al., 2019*); mGlu5 (*Werthmann et al., 2021*), Gαo-RLuc, Venus-Gγ2 (*Ayoub et al., 2009*); Flag-G$\beta_2$ (*Rajalu et al., 2015*); Myc-GB1a, HA-GB2 (*Pagano et al., 2001*); Kir3.1/Kir3.2 concatamer (*Wischmeyer et al., 1997*); PKA-R-RLuc-N, PKA-C-RLuc-C (*Stefan et al., 2007*); and SRE-FLuc (*Cheng et al., 2010*). GABA, CGP54626, forskolin, picrotoxin, and tetrodotoxin (TTX) were from Tocris Bioscience, Bristol, England.

## Peptide characterization
APP17 (Ac-DDSDVWWGGADTDYADG-NH$_2$; *Rice et al., 2019*) and sc-APP17 (acetyl-DWGADTVSGDGYDAWDD-amide) peptides were from Insight Biotechnology, London, England (>98% purity). ESI-LC-MS (Poroshell, 300 SB-C18, 2.1×75 mm, Agilent Technologies, Santa Clara, CA, USA) and RP-UPLC (Acquity, Waters Corporation, Milford, CT, USA) were used to confirm peptide mass and purity, respectively. ITC experiments were carried out in a microcalorimeter (Microcal ITC200, GE Healthcare, Chicago, IL, USA) at 25°C with a stirring speed of 600 rpm in a buffer containing 20 mM NaPi (pH 6.8), 50 mM NaCl, and 0.5 mM EDTA. For titration, APP17 or sc-APP17 (each 300 µM) were injected (first injection 0.5 µl, followed by 25 injections of 1.5 µl) into the sample cell containing purified recombinant SD1/2 protein (30 µM) (*Schwenk et al., 2016*). Control measurements of peptide versus buffer were subtracted from the peptide versus SD1/2 measurements. Data were analyzed with Microcal ITC200 Origin software, using a one-site binding model.

## Cell lines
HEK293T were directly obtained from ATCC (https://web.expasy.org/cellosaurus/CVCL_0063) and maintained in DMEM supplemented with 10% FBS (GE Healthcare) and 2% penicillin/streptomycin (Sigma-Aldrich, St Louis, MO, USA) at 37°C with 5% $CO_2$. HEK293T cells stably expressing Gαqi were a gift from the laboratory of Murim Choi (Seoul National University College of Medicine, Republic of Korea). All cell lines were authenticated using short tandem repeat analysis by Microsynth (Switzerland) and tested negative for mycoplasma contamination.

## Cell culture and transfection
HEK293T cells were transiently transfected in Opti-MEM (Gibco, Thermo Fisher Scientific) using Lipofectamine 2000 (Thermo Fisher Scientific). The total amount of transfected DNA was kept equal by supplementing with pCI plasmid DNA (Promega, Madison, WI, USA). For electrophysiological recordings, transfected cells were seeded on poly-L-lysine (Sigma-Aldrich) coated coverslips. Transfected cells were identified by their EGFP fluorescence. To establish primary cultures of hippocampal neurons, pregnant RjOrl:SWISS mice (Janvier Labs, France) were sacrificed under anesthesia by decapitation (Animal license number 1897_31476, approved by the Veterinary Office of Basel-Stadt, Switzerland). Dissected hippocampi of E17/18 embryos were collected in HBSS (Gibco, Thermo Fisher Scientific)

and dissociated with 0.25% trypsin (Invitrogen, Thermo Fisher Scientific) at 37°C for 10 min. Cells were suspended in dissection medium (MEM Eagle [Sigma-Aldrich]; 0.5% D(+)glucose; 10% horse serum [Gibco, Thermo Fisher Scientific]; 0.1% Pen-Strep [Sigma-Aldrich]) to block trypsin activity. Cells were plated on 13 mm cell culture coverslips coated with 0.01 mg/ml poly-L-lysine hydrobromide (Sigma-Aldrich) at a density of 50,000 cells/cm². Two hr after dissection, the medium was replaced with Neurobasal Medium (Gibco, Thermo Fisher Scientific) supplemented with B-27 (Gibco, Thermo Fisher Scientific) and GlutaMAX (Thermo Fisher Scientific). Primary hippocampal neurons were maintained in a humidified incubator with 5% $CO_2$ at 37°C.

## APP17-TMR binding experiments

Transfected HEK293T cells expressing Flag-GB1a and Flag-GB2 were seeded into 96-well microplates (Greiner Bio-One, Kremsmünster, Austria) at 50,000 cells/well. After 18 hr, peptides were mixed with conditioned medium at the following final concentrations: APP17-TMR (1 µM) with either APP17 (10 µM), sc-APP17 (10 µM), or PBS (vehicle); sc-APP17-TMR (1 µM) in PBS was used as a negative control. After removal of medium, peptide mixes were added to the wells and cells incubated in the dark for 1 hr at room temperature. After removal of the peptides, PBS with $MgCl_2$ and $CaCl_2$ (Sigma-Aldrich) was added to the wells. TMR fluorescence was monitored with a Spark microplate reader (Tecan Group, Männerdorf, Switzerland) using a monochromator (Excitation 544 nM, 20 nM bandwidth; detection 594 nM, 25 nm bandwidth). TMR fluorescence was determined after subtraction of the sc-APP17-TMR fluorescence measured at HEK293T cells transfected with pCI plasmid.

## FRET measurements

Single and combined labeling of SNAP- and ACP-tag were performed as described previously (*Lecat-Guillet et al., 2017*). Briefly, 24 hr after transfection, cells were incubated for 24 hr at 30°C. The medium was removed and cells were incubated with 500 nM of SNAP-Red in Tag-Lite Buffer (Perkin Elmer Cisbio) for 1 hr at 37°C. Cells were washed once and incubated with 10 mM $MgCl_2$, 1 mM DTT, 2 µM CoA-Lumi4-Tb (Perkin Elmer Cisbio) and Sfp synthase (New England Biolabs) in Tag-Lite Buffer for 1 hr at 30°C. Cells were washed three times and APP17 or scAPP17 were added either alone or together with GABA in Tag-Lite Buffer. TR-FRET measurements were performed in Greiner black 96-well plates, using a PHERAstar FS microplate reader. After excitation with a laser at 337 nm (40 flashes per well), the fluorescence was collected at 620 nm (donor signal) and 665 nm (sensitized acceptor signal). The acceptor ratio was calculated using the sensitized acceptor signal integrated over the time window (50–100 µs), divided by the sensitized acceptor signal integrated over the time window (900–1150 µs).

## BRET measurements

BRET experiments were performed as described (*Ivankova et al., 2013*; *Turecek et al., 2014*; *Dinamarca et al., 2019*). HEK293T cells were transiently transfected with Flag-GB1a, Flag-GB2, Gαo-RLuc, Gβ2, and Venus-Gγ2 plasmids with or without APP695. In order to ensure APP695 binding to GB1a/2 in trans a pool of HEK293T cells expressing APP695 was mixed with a pool of HEK293T cells expressing Flag-GB1a, Flag-GB2, Gαo-RLuc, Gβ2, and Venus-Gγ2. Transfected cells were seeded into 96-well microplates (Greiner Bio-One) at 100,000 cells/well. After 18 hr, cells were washed and coelenterazine h (5 µM, NanoLight Technologies, Prolume Ltd., Pinetop-Lakeside, AZ, USA) added for 5 min. Luminescence and fluorescence signals were alternatively recorded for a total of 845 s using a Spark microplate reader. Peptide, GABA, or CGP54626 were injected with the Spark microplate reader injection system at either 146 or 457 s. The BRET ratio was calculated as the ratio of the light emitted by Venus-Gγ2 (530–570 nm) over the light emitted by Gαo-RLuc (370–470 nm). BRET ratios were adjusted by subtracting the ratios obtained when RLuc fusion proteins were expressed alone. Each data point represents a technical quadruplicate.

## PKA assay

PKA measurements were performed as described in *Stefan et al., 2007*. HEK293T cells were transiently transfected with Flag-GB1a, Flag-GB2, PKA-Reg-RLuc-NT, and PKA-Cat-RLuc-CT with or without APP695. Transfected cells were distributed into 96-well microplates (Greiner Bio-One) at a density of 80,000 cells/well. After 42 hr, cells were washed and coelenterazine h (5 µM, NanoLight

Technologies) added for 5 min. Luminescence signals were detected for a total of 1276 s using a Spark microplate reader. To induce PKA dissociation, 1 mM forskolin was added manually at 108 s. Peptide, GABA, or CGP54626 were injected at either 529 or 905 s. Luminescence signals were adjusted to luminescence signals obtained by injecting PBS at 529 and 905 s. The luminescence was normalized to baseline luminescence. Curves were plotted after forskolin addition and the time point 71 s prior the first injection was set to 0. Each data point represents a technical quadruplicate.

## PLC and Ca$^{2+}$ mobilization assays

HEK293T cells stably expressing Gαqi were transiently transfected with Flag-GB1a, Flag-GB2, and SRE-FLuc with or without APP695. In order to ensure GB1a/2 binding of APP695 in trans a pool of HEK293T-Gαqi cells expressing APP695 was mixed with HEK293T-Gαqi cells expressing Flag-GB1a, Flag-GB2, and SRE-FLuc. Transfected cells were distributed into 96-well microplates (Greiner Bio-One) at a density of 80,000 cells/well. After 24 hr, the culture medium was replaced with Opti-MEM-GlutaMAX. Peptides were incubated in Opti-MEM-GlutaMAX for 1 hr. In presence of peptide, GB1a/2 receptors were activated with various concentrations of GABA for 15 hr. FLuc activity in lysed cells was measured using the Luciferase Assay Kit (Promega) using a Spark microplate reader. Luminescence signals were adjusted by subtracting the luminescence obtained when expressing SRE-FLuc fusion proteins alone. Ca$^{2+}$ mobilization experiments were performed as described (*Werthmann et al., 2021*). Intracellular Ca$^{2+}$ signals were detected using a Spark microplate reader at 0.85 Hz and normalized to baseline fluorescence. A monochromator was used for excitation and fluorescence collection at 485 and 530 nm, respectively. The peak of Ca$^{2+}$ mobilization was determined by the mean of four data points 3.5 s after application of 1 or 10 μM APP17, sc-APP17, GABA, or (S)-3,5-DHPG. GB1a/2 receptor expression in HEK293T cells was validated using immunocytochemistry. HEK293T cells were fixed with 4% PFA (Sigma) + 4% sucrose (Sigma) in PBS (Life Technologies) and permeabilized with PBS + 0.2% Triton X-100 (Sigma) + 10% normal donkey serum (Abcam). After washing the cells with PBS, either primary mouse anti-GB1 (Abcam) or guinea pig anti-GB2 (Synaptic Systems) antibody was incubated overnight in the dark at 4°C. The next day, cells were washed with PBS and either secondary Alexa Fluor 647 donkey anti-guinea pig IgG (Life Technologies) or secondary Alexa Fluor 647 donkey anti-mouse IgG (Life Technologies) antibody was incubated for 45 min at room temperature in the dark. Fluorescence was measured using a Spark microplate reader. A monochromator was used for excitation at 640 nm and fluorescence collection at 685 nm.

## Quantification of synaptic GBRs

Cultured hippocampal neurons (DIV14) on glass coverslips were treated with 5 μM APP17 or sc-APP17 for 1 hr, fixed with PBS (Life Technologies) containing 4% PFA (Sigma) and 4% sucrose (Sigma) and permeabilized with PBS containing 0.2% Triton X-100 (Sigma) and 10% normal donkey serum (Abcam). Neurons were washed with PBS and blocked with PBS containing 10% normal donkey serum (Abcam). Primary guinea pig anti-GB2 (Synaptic Systems), rabbit anti-vGluT1 (Synaptic Systems), and mouse anti-PSD-95 (Synaptic Systems) antibodies were incubated for 2 hr at room temperature. For GBR quantification, we used a GB2 antibody because the available GB1a antibody may compete with APP17 for binding to SD1. Neurons were washed with PBS and incubated with secondary Alexa Fluor 488 donkey anti-guinea pig IgG (Jackson ImmunoResearch), Alexa Fluor 555 donkey anti-mouse IgG (Life Technologies), and Alexa Fluor 647 donkey anti-rabbit IgG (Jackson ImmunoResearch) antibodies for 45 min in the dark. Neurons were washed with PBS, air-dried, and mounted with fluorescent mounting medium (Dako). Images were captured using a Zeiss LSM-700 confocal microscope with a Plan-Apochromat 63×/NA 1.40 oil DIC, using Zen 2010 software. Fiji (ImageJ, version 1.53t) was used for image analysis. Images were z-stacked using maximum intensity. A binary mask was created for vGluT1 and PSD-95 overlap detection. Particles that contained both vGluT1 and PSD-95 were considered synapses. The GB2 fluorescence in z-stacked (maximum intensity) images was quantified at these synapses. The average GB2 fluorescence at synapses for each image represents one data point in *Figure 6B*.

## Electrophysiology

Neuronal cultures and HEK293T cells were prepared as described (*Dinamarca et al., 2019*). Coverslips with hippocampal neurons (DIV 12–15) or HEK293T cells were transferred to a chamber containing

a low-$K^+$ bath solution (in mM): 145 NaCl, 4 KCl, 5 HEPES, 5.5 D-glucose, 1 $MgCl_2$, and 1.8 $CaCl_2$ (pH 7.4 adjusted with NaOH). Recordings were performed at room temperature using borosilicate pipettes of 3–5 MΩ resistance tips, filled with K-gluconate-based pipette solution (in mM): 150 K-gluconate, 1.1 EGTA, 10 HEPES, 10 Tris-phosphocreatine, 0.3 NaGTP, and 4 MgATP (pH 7.2 adjusted with KOH). Upon achieving whole-cell access, cells were held in voltage-clamp mode at −70 mV (with no correction for liquid junction potential) and baclofen-induced $K^+$ currents were induced in a high-$K^+$ bath solution (in mM): 120 NaCl, 25 KCl, 5 HEPES, 5.5 D-glucose, 1 $MgCl_2$, and 1.8 $CaCl_2$ (pH 7.4 adjusted with NaOH). Whole-cell patch-clamp recordings from visually identified CA1 pyramidal cells in acute hippocampal slices of juvenile male and female C57BL/6JRj mice (Janvier Labs, France) were performed as described (*Vigot et al., 2006*) (animal license number 1897_31476, approved by the Veterinary Office of Basel-Stadt, Switzerland). Schaffer collaterals were stimulated at 0.1 Hz with brief current pulses via bipolar Pt/Ir wires. Evoked EPSCs were recorded at –60 mV with a $Cs^+$-based intracellular solution. Baclofen and peptides were bath applied in standard ACSF at room temperature. We recorded mEPSCs in primary neuronal cultures established from embryonic (E15–17) murine hippocampus at DIV13–16 in the presence of 0.5 µM TTX at room temperature (*Dinamarca et al., 2019*). APP17 (250 nM) and (R)-baclofen (30 µM) were applied via superfusion. Template matching with pClamp 10 software (Molecular Devices) was used to detect individual mEPSCs.

## [$^{35}$S]GTPγS binding

Preparation of mouse brain membranes was as described earlier (*Olpe et al., 1990*). Briefly, 8-week-old male C57BL/6JRj mice (Janvier Labs, France) were decapitated under isoflurane anesthesia (animal license number 1897_31476, approved by the Veterinary Office of Basel-Stadt, Switzerland). The brains were removed, washed in ice-cold PBS, and homogenized in 10 volumes of ice-cold 0.32 M sucrose, containing 4 mM HEPES, 1 mM EDTA, and 1 mM EGTA, using a glass-teflon homogenizer. Debris was removed at 1000 × *g* for 10 min and membranes were centrifuged at 26,000 × *g* for 15 min. The pellet was osmotically shocked by re-suspension in a 10-fold volume of ice-cold $H_2O$ and kept on ice for 1 hr. The suspension was centrifuged at 38,000 × *g* for 20 min and re-suspended in a threefold volume of $H_2O$. Aliquots were frozen in liquid nitrogen and stored at −20°C for 48 hr. After thawing at room temperature, a sevenfold volume of Krebs-Henseleit (KH) buffer (pH 7.4) was added, containing 20 mM Tris-HCl, 118 mM NaCl, 5.6 mM glucose, 4.7 mM KCl, 1.8 mM $CaCl_2$, 1.2 mM $KH_2PO_4$, and 1.2 mM $MgSO_4$. Membranes were washed three times by centrifugation at 26,000 × *g* for 15 min, followed by re-suspension in KH buffer. The final pellet was re-suspended in a fivefold volume of KH buffer. Aliquots of 2 ml were frozen and stored at −80°C until the day of the experiment. On the day of the experiment, frozen membranes were thawed, homogenized in 10 ml ice-cold assay buffer I containing 50 mM Tris-HCl buffer (pH 7.7); 10 mM $MgCl_2$, 1.8 mM $CaCl_2$, 100 mM NaCl, and centrifuged at 20,000 × *g* for 15 min. The pellet was re-suspended in the same volume of cold buffer and centrifuged twice as above with 30 min of incubation on ice in between the centrifugation steps. The resulting pellet was re-suspended in 150 µl of assay buffer II (per point) containing 50 mM Tris-HCl buffer (pH 7.7); 10 mM $MgCl_2$, 1.8 mM $CaCl_2$, 100 mM NaCl, 30 µM guanosine 5'-diphosphate (Sigma-Aldrich) and 20 µg of total membrane protein. To this, 8 µM of the APP17 peptide was added in 25 µl of phosphate buffer (50 mM sodium phosphate, pH 6.8, 50 mM NaCl, for +APP17) or 25 µl of phosphate buffer alone (for –APP17) and incubated for 30 min. The reaction was started by adding various concentrations of GABA and 0.2 nM of [$^{35}$S]GTPγS (PerkinElmer, Waltham, MA, USA) in a final volume of 200 µl per point and assayed as described earlier (*Rajalu et al., 2015*). Non-specific binding was measured in the presence of unlabeled GTPγS (10 µM; Sigma-Aldrich). The reagents were incubated for 1 hr at room temperature in 96-well polypropylene microplates (Greiner Bio-One) with mild shaking. They were subsequently filtered using 96-well Whatman GF/C glass fiber filters (PerkinElmer), pre-soaked in assay buffer, using a Filtermate cell harvester (PerkinElmer). After four washes with assay buffer, the Whatman filter fibers were dried for 2 hr at 50°C. Fifty µl of scintillation fluid (MicroScint-20; PerkinElmer) was added, the plates were shaken for 1 hr and thereafter counted using a Packard TopCount NXT (PerkinElmer).

## In vivo two-photon $Ca^{2+}$ imaging of auditory cortex

$Ca^{2+}$ imaging experiments were approved by the Veterinary Office of Basel-Stadt, Switzerland (animal license number 3004_34045). We crossed Ai95(RCL-GCaMP6f)-D mice (*Madisen et al., 2015*)

(RRID:IMSR_JAX:028865) with Nex-Cre mice (**Goebbels et al., 2006**) to obtain GCaMP6f expression in cortical neurons. Nine- to 12-week-old male mice were anesthetized with isoflurane (4% induction, 1.5–2.5% surgery, 1% optical imaging); 3.2 mg/kg dexamethasone was administered intraperitoneally 48, 24, and 1 hr prior to surgery to prevent brain swelling. Bupivacaine/lidocaine (0.01/0.04 mg) was injected subcutaneously for analgesia. An imaging chamber was created with cement above right auditory cortex (rACx) and a post fixed on the skull of the left hemisphere. A craniotomy (1.5×1.5 mm$^2$) above rACx was carefully opened and duratomy was performed. For optical imaging, the post was stably fixed on a stage and the head tilted 30° for optimal access to the rACx. The imaging chamber was perfused at 1 ml/min using a peristaltic pump. All solutions were kept at 37°C for 1 hr prior to perfusion. Two-photon Ca$^{2+}$ imaging periods (5 min each) started 45 min after the first ACSF perfusion, 60 min after sc-APP17, second ACSF, APP17 and third ACSF perfusion and 15 min after baclofen perfusion. Ca$^{2+}$ transients of neuronal cell bodies in the upper layers of rACx (focal depth: 150–250 μm) were recorded with a two-photon microscope (INSS) equipped with an 8 kHz resonant scanner. Images were acquired with a PXI-1073-based data acquisition system (NI) through a Nikon 16× objective (0.8 NA), at 30 Hz within a 500×500 μm$^2$ field of view (512×512 pixels). The wavelength to excite GCaMP6f was 940 nm (Chameleon Vision-S, Coherent). We used the optical imaging control and data acquisition software ScanImage 5.7 (**Pologruto et al., 2003**). Correction of Z-drift and motion artifacts, detection of neuronal cell bodies, and extraction of Ca$^{2+}$ signals were with the Python 3-based image processing pipeline suite2p (**Pachitariu et al., 2017**). Data analysis, visualization, and statistics were performed using custom-written MATLAB scripts (**Reinartz, 2023**). Automated detection of Ca$^{2+}$ transients was with an adapted algorithm (**Sorensen et al., 2017**). First, the fluorescence signal was corrected by the median filtered data removing slow trends. The detection threshold was set to 2.5 times the standard deviation and a minimum peak width of 3 data points above threshold to remove fast artifacts. $\Delta F/F$ was calculated as $(F-F_0)/F_0$, where $F_0$ is the mean fluorescence across all detected neurons in a given condition. The rates of Ca$^{2+}$ transients representing the recorded neuronal populations for each condition were plotted as the empirical cumulative distribution function with 95% confidence intervals.

## Statistical analysis

Data was analyzed with GraphPad Prism version 8 (GraphPad, San Diego, CA, USA) if not indicated otherwise. Sample size in all experiments was based on those of similar experiments in previous studies. Samples were randomly allocated to the experimental groups. Confounding effects in the cell-based assays (**Figures 2–5**) were minimized by rotating the order of cell plating. Blinding was not performed for any experiment. Individual datasets were tested for normality with the Shapiro-Wilk or D'Agostino-Pearson test (for n≥8). For the datasets obtained with the cell-based assays (**Figures 2–5**) outliers were identified using the ROUT method with Q=1%. For all other experiments, no data was excluded. Statistical significance of datasets against 0 or 100 was assessed by one-sample t-test. For non-normal distribution, the non-parametric one-sample Wilcoxon test was used. Statistical significance between two groups containing one variable was assessed by Student's t-test. Statistical significance between three or more groups containing one variable was assessed by ordinary or paired one-way ANOVA with Holm-Sidak's multiple comparisons test. For non-normal distribution, the nonparametric Friedman test with Dunn's multiple comparisons test was used. Statistical significance between groups containing two variables was assessed by ordinary two-way ANOVA with Sidak's multiple comparisons test. Statistical significance between dose-response curves was assessed by extra sum-of-squares F test of non-linear regression curve fits. p-Values < 0.05 were considered significant. Data are presented as mean ± standard error of mean (SEM) or mean ± standard deviation (SD) as indicated in the figure legends.

## Acknowledgements

We thank M Hasegawa and S Kandler for helpful comments regarding calcium imaging technology.

# Additional information

## Competing interests

Kristian Strømgaard: is a co-founder and a part time employee of Avilex Pharma. Bernhard Bettler: is a member of the scientific advisory board of Addex Therapeutics, Geneva. The other authors declare that no competing interests exist.

## Funding

| Funder | Grant reference number | Author |
|---|---|---|
| Schweizerischer Nationalfonds zur Förderung der Wissenschaftlichen Forschung | 31003A-152970 | Bernhard Bettler |
| Brain and Behavior Research Foundation | NARSAD Young Investigator Grant | Sebastian Reinartz |
| Brain and Behavior Research Foundation | 30389 | Sebastian Reinartz |
| Deutsche Forschungsgemeinschaft | 439189341 | Bernd Fakler |
| Deutsche Forschungsgemeinschaft | 446245862 | Bernd Fakler |
| Deutsche Forschungsgemeinschaft | 239283807 | Bernd Fakler |

The funders had no role in study design, data collection and interpretation, or the decision to submit the work for publication.

## Author contributions

Pascal Dominic Rem, Conceptualization, Formal analysis, Investigation, Writing – original draft; Vita Sereikaite, Diego Fernández-Fernández, Sebastian Reinartz, Daniel Ulrich, Thorsten Fritzius, Luca Trovo, Salomé Roux, Ziyang Chen, Formal analysis, Investigation; Philippe Rondard, Conceptualization, Formal analysis; Jean-Philippe Pin, Bernd Fakler, Tania Rinaldi Barkat, Kristian Strømgaard, Conceptualization; Jochen Schwenk, Conceptualization, Investigation; Martin Gassmann, Conceptualization, Investigation, Writing – review and editing; Bernhard Bettler, Conceptualization, Funding acquisition, Writing – original draft, Writing – review and editing

## Author ORCIDs

Diego Fernández-Fernández ⓘ http://orcid.org/0000-0003-1431-3705
Thorsten Fritzius ⓘ http://orcid.org/0000-0002-3597-6623
Salomé Roux ⓘ http://orcid.org/0000-0002-6106-4863
Philippe Rondard ⓘ http://orcid.org/0000-0003-1134-2738
Jochen Schwenk ⓘ http://orcid.org/0000-0003-3664-9795
Bernd Fakler ⓘ http://orcid.org/0000-0001-7264-6423
Tania Rinaldi Barkat ⓘ http://orcid.org/0000-0001-8650-0986
Kristian Strømgaard ⓘ http://orcid.org/0000-0003-2206-4737
Bernhard Bettler ⓘ http://orcid.org/0000-0003-0842-8207

## Ethics

All animal experiments were approved by the veterinary office of the canton of Basel-Stadt, Switzerland (animal license numbers: 1897_31476 and 3004_34045).

## Decision letter and Author response

Decision letter https://doi.org/10.7554/eLife.82082.sa1
Author response https://doi.org/10.7554/eLife.82082.sa2

## Additional files

### Supplementary files
- MDAR checklist
- Reporting standard 1. ARRIVE guidelines 2.0 checklist.

### Data availability
For all figures, numerical data that are represented in graphs are provided as source data excel files.

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
