## [Editor Report]

The manuscript is very relevant to the field of Alzheimer's disease and the signal transduction mechanisms of GABA_B_ receptors. It is a rigorous study that addresses the physiological function of peptide isoforms of APP vis-a-vie G protein-coupled GABA_B_ receptors. The manuscript presents strong experimental evidence of this interaction but also concludes that although an APP peptide, APP17, binds a GABA_B_ receptor, there is a lack of clearly demonstrable functional effects.

---

## [Decision Letter]

**Decision letter after peer review:**

Thank you for submitting your article "Soluble amyloid-β precursor peptide does not regulate GABAB receptor activity" for consideration by *eLife*. Your article has been reviewed by 2 peer reviewers, and the evaluation has been overseen by a Reviewing Editor and John Huguenard as the Senior Editor. The following individuals involved in the review of your submission have agreed to reveal their identity: Shiva K. Tyagarajan (Reviewer #1); Christophe Mulle (Reviewer #2). We apologize for the extremely lengthy review process of your manuscript. A long delay in obtaining a review held up the process.

The reviewers have discussed their reviews with one another, and the Reviewing Editor has drafted this to help you prepare a revised submission. We appreciate your understanding of the editorial process. Thank you for your patience in the review of your study.

Essential revisions:

The two reviewers both agreed the study of sAPP and GABA receptors had considerable merit and importance for the fields of GABA biology and Alzheimer's research. The referees felt the study could be strengthened with further attention to sAPP on GABAB receptors, such as the effect 0f APP17 on mEPSC frequency in primary hippocampal neurons. This was brought up by reviewer #2 and endorsed by the first reviewer. Therefore we wish to ask for a revised manuscript to address this recommendation. Several revisions to look at ca^2+^, lipid signaling, and the role of microglia were also suggested, but are of lower priority depending on the likelihood of meaningful results.

*Reviewer #1 (Recommendations for the authors):*

I would recommend that authors test some ideas to identify the biological relevance of APP interaction with GABABR with such high affinity. My suggested experiments would be as follows:

1. In 2018 Nicholson et al., reported that prolonged exposure to benzodiazepine activates GPCR PLC δ /calcineurin pathway leading to GABAAR internalization from synaptic sites. It will be worthwhile to measure how the surface/synaptic expression of GABABRs is influenced by APP17 binding.

2. Is lipid signalling from the plasma membrane facilitating downstream effects after APP17 binding to GB1a?

3. What happens to intracellular ca^2+^ levels after APP17 binding to GABABRs? the GCaMP in vivo measurements might not be sensitive to measure internal ca^2+^ release. In HEK293T cells, authors can measure internal changes in ca^2+^ using a calcium-sensitive dye after APP17 application.

4. GABAB receptor function on microglia facilitates synapse pruning during development (Favuzzi et al., 2021), we reported GABA action on microglia regulates its activation status (Cramer et al., 2022). Is sAPP binding to GABAB receptor influencing microglia and not neuronal function? This can be tested using the microglia cell line BV2 or primary microglia.

Cannabinoid pathway activation in microglia/neurons could provide a link to lipid signalling.

*Reviewer #2 (Recommendations for the authors):*

The authors provide an extensive range of techniques used to invalidate in a well-controlled manner the claim that sAPP has an agonistic (or PAM) activity on GABAB receptors.

It is therefore unclear why the authors do not attempt to replicate precisely some of the electrophysiological experiments reported in the Rice et al. paper. This is particularly the case for the test of APP17 on mEPSC frequency in cultured mouse hippocampal neurons, which is a straightforward experiment. Either the results differ from that of the Rice paper, or the authors find indeed a decrease in mEPSC frequency, and this may relate to a mechanism independent of sAPP binding to GABAB receptors. This would be important information and related discussion.

---

## [Author Response]

Reviewer #1 (Recommendations for the authors):I would recommend that authors test some ideas to identify the biological relevance of APP interaction with GABABR with such high affinity.

Binding of APP to GABAB receptors is important for axonal trafficking and surface stability of GABAB receptors (Dinamarca et al., 2019), which provides a biological function for the APP/GABAB receptor interaction. Our analysis shows that binding of APP17 or sAPP to GABAB receptors is without functional consequences in a comprehensive set of in vitro and in vivo assays. Moreover, we now additionally show that APP17 does not influence synaptic localization of GABAB receptors (Figure 6).

1. In 2018 Nicholson et al., reported that prolonged exposure to benzodiazepine activates GPCR PLC δ /calcineurin pathway leading to GABAAR internalization from synaptic sites. It will be worthwhile to measure how the surface/synaptic expression of GABABRs is influenced by APP17 binding.

Full-length APP stabilizes GABAB receptors at the cell surface and prevents receptor degradation (Dinamarca et al., 2019). It is therefore conceivable that APP17 influences receptor expression and degradation. We now show that prolonged exposure of cultured hippocampal neurons to APP17 does not significantly influence synaptic GABAB receptor expression (Figure 6).

P9. “APP17 does not influence synaptic GBR localization

It is conceivable that prolonged exposure of cultured hippocampal neurons to APP17 affects synaptic GBR levels, for example by stabilizing GB1a/2 receptors at the cell surface or displacing APP from GB1a/2 receptors (Dinamarca et al., 2019; Hannan et al., 2012). Exposure of cultured hippocampal neurons to 5 μM APP17 or sc-APP17 for one hour did not result in a significant difference in GB2 protein levels at glutamatergic synapses, which were identified by vGluT1 and PSD95 immunostaining (Figure 6). The functional effects of APP17 previously observed in cultured neurons (Rice et al., 2019) are therefore not caused by altered GBR levels at synapses.”

P13. “Consistent with a lack of APP17 effects on GBRs, we found no evidence that prolonged application of APP17 alters synaptic GBR levels.”

P19. “Quantification of synaptic GBRs

Cultured hippocampal neurons (DIV14) on glass coverslips were treated with 5 µM APP17 or sc-APP17 for 1 hr, fixed with PBS (Life Technologies) containing 4% PFA (Σ) and 4% sucrose (Σ) and permeabilized with PBS containing 0.2% Triton X-100 (Σ) and 10% normal donkey serum (Abcam). Neurons were washed with PBS and blocked with PBS containing 10% normal donkey serum (Abcam). Primary guinea pig anti-GB2 (Synaptic Systems), rabbit anti-vGluT1 (Synaptic Systems) and mouse anti-PSD-95 (Synaptic Systems) antibodies were incubated for 2 hrs at room temperature. For GBR quantification we used a GB2 antibody because the available GB1a antibody may compete with APP17 for binding to SD1. Neurons were washed with PBS and incubated with secondary Alexa Fluor 488 donkey anti-guinea pig IgG (Jackson ImmunoResearch), Alexa Fluor 555 donkey anti-mouse IgG (Life Technologies) and Alexa Fluor 647 donkey anti-rabbit IgG (Jackson ImmunoResearch) antibodies for 45 min in the dark. Neurons were washed with PBS, air dried and mounted with fluorescent mounting medium (Dako). Images were captured using a Zeiss LSM-700 confocal microscope with a Plan-Apochromat 63 × /NA 1.40 oil DIC, using Zen 2010 software. Fiji (ImageJ, version 1.53t) was used for image analysis. Images were Z-stacked using maximum intensity. A binary mask was created for vGluT1 and PSD-95 overlap detection. Particles that contained both vGluT1 and PSD-95 were considered synapses. The GB2 fluorescence in z-stacked (maximum intensity) images was quantified at these synapses. The average GB2 fluorescence at synapses for each image represents one data point in Figure 6B.”

P36. “Figure 6. APP17 does not influence GBR levels at glutamatergic synapses. (A) Images of dendrites of cultured hippocampal neurons (DIV14) exposed to 5 µM APP17 or sc-APP17 for 1 hour. Immunofluorescence labeling of GB2 (Alexa Fluor 488), vGluT1 (Alexa Fluor 647), PSD95 (Alexa Fluor 555) and their overlay (merged) is shown. Dashed lines indicate examples of vGluT1/PSD95 synapses. Scale bar: 5 µM. (B) Bar graphs showing GB2 (Alexa Fluo 488) immunofluorescence at vGluT1/PSD95 synapses. Data are mean ± SEM. The number of analyzed images from three independent preparations is indicated. ns = not significant, Mann Whitney test (non-parametric). Source file containing GB2 immunofluorescence data is available in Figure 6 – source data 1.”

2. Is lipid signalling from the plasma membrane facilitating downstream effects after APP17 binding to GB1a?

Our data show the absence of “downstream effects after APP17 binding to GB1a”, using a broad array of classical in vitro functional assays for GABAB receptors. In response to the reviewer’s comment, we additionally found no evidence for APP17 acting as a biased ligand favoring signaling through Gαq instead of Gαo/i. Likewise, APP17 neither influenced synaptic transmission nor neuronal activity. Because of the complete absence of functional effects of APP17 at GABAB receptors in vitro and in vivo, there is no clear rationale for exploring a “facilitation” of GABAB receptor signaling by lipid signaling.

3. What happens to intracellular ca^2+^ levels after APP17 binding to GABABRs? the GCaMP in vivo measurements might not be sensitive to measure internal ca^2+^ release. In HEK293T cells, authors can measure internal changes in ca^2+^ using a calcium-sensitive dye after APP17 application.

In our experiments (Figure 10) we used the GCaMP sensor for monitoring neuronal activity in mice and not for imaging putative GABAB receptor-mediated increases in intracellular ca^2+^. GABAB receptors are well-known to couple to Gαi/o-type G proteins, which do not activate PLC and lead to increases in intracellular ca^2+^. However, it has been proposed that GABAB receptors couple to Gαq and increase intracellular ca^2+^ in some neuronal populations (Karls and Mynlieff, 2015). In response to the reviewer’s comment we tested whether APP17 biases GABAB receptor signaling towards Gαq and studied a possible increase in intracellular ca^2+^ in response to APP17 (Figure 5 —figure supplement 2). Our results show that APP17 does not increase intracellular ca^2+^ in HEK293T cells expressing GABAB receptors.

P8. “APP17 does not bias GB1a/2 signaling towards Gαq

GBRs are classical Gαo/i-coupled receptors that normally lack the ability to mobilize intracellular ca^2+^ (Pin and Bettler, 2016). However, it has been proposed that in some neurons GBRs increase intracellular ca^2+^ by activating Gαq and phospholipase C (Karls and Mynlieff, 2015). It is possible that APP17 is a biased ligand preferentially stabilizing a conformational state of GBRs that elicits signaling through Gαq (Slosky et al., 2021). APP17 did not activate PLC in the SRE-luciferase assay (Figure 5). However, high levels of exogenous Gαqi in this assay may interfere with signaling through endogenous Gαq. We therefore analyzed APP17 activity at GB1a/2 receptors in a ca^2+^ mobilization assay (Werthmann et al., 2021). HEK293T cells expressing GB1a/2 or control mGlu5 receptors were loaded with the ca^2+^ indicator Calcium 6 and stimulated with 1 or 10 μM APP17, GABA or (S)-3,5-DHPG (Figure 5 —figure supplement 2). Application of APP17 or GABA did not induce significant ca^2+^ mobilization in cells expressing GB1a/2 receptors (Figure 5 —figure supplement 2A,B). Expression of GB1a/2 receptors in HEK2093T cells was confirmed by immunostaining for GB1 and GB2 (Figure 5 —figure supplement 2C). Application of the selective group I mGlu receptor agonist DHPG to cells expressing mGlu5 receptors induced a concentration-dependent increase in intracellular ca^2+^ (Figure 5 —figure supplement 2A,B), as expected (Werthmann et al., 2021). These experiments indicate that APP17 does not elicit Gαq signaling at GB1a/2 receptors in HEK293T cells.”

P13. “It is well established that GBRs normally signal through Gαo/i-type G proteins (Pin and Bettler, 2016). However, Gαq signaling was proposed as an alternative signaling mode of GBRs in select neuronal populations (Karls and Mynlieff, 2015). We therefore considered the possibility that APP17 acts as a biased ligand favoring Gαq over Gαo/i signaling However, we found no evidence that APP17 mobilizes intracellular ca^2+^ in transfected HEK293T cells expressing GB1a/2.”

P18. “ca^2+^ mobilization experiments were performed as described (Werthmann et al., 2021). Intracellular ca^2+^ signals were detected using a Spark microplate reader at 0.85 Hz and normalized to baseline fluorescence. A monochromator was used for excitation and fluorescence collection at 485 nm and 530 nm, respectively. The peak of ca^2+^ mobilization was determined by the mean of four data points 3.5 sec after application of 1 µM or 10 µM APP17, sc-APP17, GABA or (S)-3,5-DHPG. GB1a/2 receptor expression in HEK293T cells was validated using immunocytochemistry. HEK293T cells were fixed with 4% PFA (Σ) + 4% sucrose (Σ) in PBS (Life Technologies) and permeabilized with PBS + 0.2% Triton X-100 (Σ) + 10% normal donkey serum (Abcam). After washing the cells with PBS, either primary mouse anti-GB1 (Abcam) or guinea pig anti-GB2 (Synaptic Systems) antibody was incubated overnight in the dark at 4°C. The next day, cells were washed with PBS and either secondary Alexa Fluor 647 donkey anti-guinea pig IgG (Life Technologies) or secondary Alexa Fluor 647 donkey anti-mouse IgG (Life Technologies) antibody was incubated for 45 min at room temperature in the dark. Fluorescence was measured using a Spark microplate reader. A monochromator was used for excitation at 640 nm and fluorescence collection at 685 nm.”

P40. “Figure 5 —figure supplement 2. APP17 does not increase intracellular ca^2+^ in HEK293T cells expressing GB1a/2 receptors. (A) APP17, sc-APP17 and GABA at 1 µM or 10 µM did not increase intracellular ca^2+^ in HEK293T cells expressing GB1a/2 receptors. Cells were loaded with the ca^2+^ indicator Calcium 6. Control experiments with (S)-3,5-DHPG at 1 µM and 10 µM showed concentration-dependent ca^2+^ increases in cells expressing mGlu5. The fluorescence intensity relative to basal levels (F/F0) over time is depicted. (B) Bar graphs showing maximal fluorescence intensity changes in response to drug application to GB1a/2 or mGlu5 expressing cells. (C) Bar graphs showing GB1 and GB2 fluorescence levels (Alexa Fluor 647) in control or GB1a/2 receptor-expressing HEK293T cells. Data are mean ± SEM. The number of independent transfections is indicated. *p < 0.05 One sample Wilcoxon test (non-parametric) against 0 (B), ****p < 0.0001, unpaired t-test (C). Source file containing ca^2+^ responses and GB1 and GB2 immunofluorescence data is available in Figure 5 —figure supplement 2 – source data 1.”

4. GABAB receptor function on microglia facilitates synapse pruning during development (Favuzzi et al., 2021), we reported GABA action on microglia regulates its activation status (Cramer et al., 2022). Is sAPP binding to GABAB receptor influencing microglia and not neuronal function? This can be tested using the microglia cell line BV2 or primary microglia.

The work by Favuzzi et al. (2021) describes an effect of GABAB receptors in a subset of microglia on a long-term developmental wiring process, which cannot be studied in a microglia cell line or primary microglia. The work by Cramer et al. (2022) does not directly implicate GABAB receptors in the regulation of microglia activation. Most importantly, however, neither the above studies nor other studies suggest a non-canonical GABAB receptor signaling in microglia. Therefore, there is no rationale for why APP17 should specifically activate a non-canonical GABAB receptor signaling in microglia. Moreover, APP17 effects on non-neuronal cells would likely influence neuronal activity in anesthetized mice, which we show remains unchanged in the presence of APP17 (Figure 10). In the absence of a clear rationale, we did not engage into speculative/exploratory experiments with microglia.

Reviewer #2 (Recommendations for the authors):The authors provide an extensive range of techniques used to invalidate in a well-controlled manner the claim that sAPP has an agonistic (or PAM) activity on GABAB receptors.It is therefore unclear why the authors do not attempt to replicate precisely some of the electrophysiological experiments reported in the Rice et al. paper. This is particularly the case for the test of APP17 on mEPSC frequency in cultured mouse hippocampal neurons, which is a straightforward experiment. Either the results differ from that of the Rice paper, or the authors find indeed a decrease in mEPSC frequency, and this may relate to a mechanism independent of sAPP binding to GABAB receptors. This would be important information and related discussion.

We found that APP17 does not influence the mEPSC frequency in cultured mouse hippocampal neurons, consistent with a lack of APP17 effect on evoked EPSCs in acute hippocampal slices (Figure 9).

P2. “Moreover, APP17 did not regulate synaptic GBR localization, GBR-activated K^+^ currents, neurotransmitter release or neuronal activity in vitro or in vivo.”

P4. “Likewise, APP17 did not influence K^+^ currents or mEPSC frequencies in cultured hippocampal neurons, reduce the amplitude of evoked EPSCs (eEPSCs) in hippocampal slices or modulate neuronal activity in living mice.”

P10. “Since our experiments showed no functional effects of APP17 at GBRs, we sought to replicate the reported decrease in mEPSC frequency after APP17 application to cultured hippocampal neurons (Rice et al., 2019). Using the same incubation time and concentration of APP17 (Rice et al., 2019), we were unable to detect a significant effect of APP17 on the mEPSC frequency of cultured hippocampal neurons. Subsequent application of baclofen to the same neurons resulted in a significantly reduced mEPSC frequency (Figure 9A,B).”

P13. “APP17 also did not regulate native GBRs in experiments assessing (1) G protein activation in brain membranes, (2) activation of K^+^ currents in cultured neurons, (3) mEPSC frequency in cultured neurons, (4) eEPSC amplitude modulation in acute hippocampal slices and, (5) neuronal activity in anaesthetized mice.”

P20. “We recorded mEPSCs in primary neuronal cultures established from embryonic (E15-17) murine hippocampus at DIV13-16 in the presence of 0.5 μM TTX at room temperature (Dinamarca et al., 2019). APP17 (250 nM) and (R)-baclofen (30 μM) were applied via superfusion. Template matching with pClamp 10 software (Molecular Devices) was used to detect individual mEPSCs.”

P37. “Figure 9. APP17 does not influence excitatory synaptic transmission. (A) Sample mEPSCs at baseline and in the presence of 0.25 µM APP17 or 30 µM baclofen (bac) recorded from cultured hippocampal neurons (DIV13-16). (B) Bar graphs showing the mEPSC frequency normalized to baseline in the presence of APP17 (blue) or baclofen (red). The number of recorded neurons is indicated. ns = not significant; ****p<0.0001, paired students t-test; ***p<0.001, one sample t-test against 100.”